# Attention-Guided Masking and Neighbor-Informed Reconstruction for Tabular Anomaly Detection

## Abstract

One-class classification for tabular anomaly detection remains challenging due to the scarcity of labeled anomalies and the absence of explicit structural relationships among samples. Existing approaches have largely focused on intra-instance modeling, such as masking or reconstruction, while inter-sample modeling has received comparatively little attention. We propose AGNI (Attention-Guided Masking and Neighbor-Informed Reconstruction), a self-supervised framework that reimagines attention as a dual supervisory signal unifying these two perspectives. Specifically, Attention-Guided Masking leverages attention to identify and hide salient features, enforcing the learning of fine-grained intra-instance dependencies. At the same time, Neighbor-Informed Reconstruction repurposes the same attention scores to retrieve structurally similar neighbors, whose representations provide contextual support during reconstruction. By tightly coupling intra-instance and inter-sample objectives within a single attention space, AGNI transforms attention from a representational tool into a coordinating structural signal. Extensive experiments on 47 real-world datasets from ADBench demonstrate that AGNI achieves the best overall ranking among 15 classical and deep-learning baseline. Code is available in the supplementary material.

Anomaly detection in tabular data plays a crucial role in diverse applications such as fraud detection, healthcare monitoring, and industrial process control (Ahmad et al., 2021; Fernando et al., 2021; Ye et al., 2023a; Al-Hashedi & Magalingam, 2021). In these domains, labeled anomalies are scarce, ambiguous, or costly to obtain, making one-class classification (OCC) a natural and widely adopted formulation Ruff et al. (2021); Ye et al. (2023b); Chandola et al. (2009); Guo et al. (2023). The central challenge in this setting is to accurately capture the distribution of normal data such that anomalies can be identified as meaningful deviations (Yin et al., 2024; Shenkar & Wolf, 2022). However, tabular data presents unique difficulties due to the heterogeneity of feature types and the lack of inherent structural relationships, which stand in sharp contrast to the spatial or sequential structures found in images and text.

Most prior work has therefore concentrated on intra-instance modeling—for example, through feature masking or reconstruction tasks—to capture dependencies within each sample (Yin et al., 2024). By contrast, inter-sample modeling, which has played a central role in anomaly detection for other modalities such as images (Gong et al., 2019), time series (Audibert et al., 2020), and graphs (Fan et al., 2020), has received relatively little attention in the tabular setting. We attribute this scarcity to the absence of explicit structural relationships, which makes it difficult to define meaningful similarity across samples (Somepalli et al., 2021). Yet inter-sample modeling is essential, as it provides complementary structural signals that help define normality beyond individual feature interactions, thereby improving robustness against subtle or context-dependent anomalies. Only recently have a handful of studies explored retrieval-based approaches for tabular data, but these remain limited by their reliance on naive embedding similarity (Thimonier et al., 2024).

As a result, intra-instance and inter-sample perspectives have remained largely isolated. Existing attention-based models for tabular data (e.g., SAINT (Somepalli et al., 2021), TabTransformer (Huang et al., 2020)) further reinforce this separation, as they employ attention only as a representational mechanism without turning it into a learning signal. We argue that a promising di-

rection is to develop a unified supervisory source that can actively coordinate both perspectives—a possibility that, to the best of our knowledge, remains unexplored in tabular anomaly detection.

In this work, we present AGNI (Attention-Guided Masking and Neighbor-Informed Reconstruction), a self-supervised framework that reimagines attention not merely as an encoding tool but as a dual supervisory signal for both feature masking and neighbor retrieval. Specifically, AGNI (i) identifies and masks structurally salient features to enforce the learning of fine-grained intra-instance dependencies, and (ii) retrieves attention-similar neighbors whose representations provide contextual support during reconstruction. By tightly coupling these objectives within a shared attention space, AGNI transforms attention into a coordinating structural signal that unifies intra- and inter-instance modeling.

Beyond empirical gains, AGNI also reveals a distinctive qualitative phenomenon: as we later demonstrate in t-SNE visualizations (Figure 4), normal samples tend to retrieve compact neighborhoods, whereas anomalous samples yield scattered neighbors. This behavior provides interpretable evidence of AGNI's structural novelty and highlights how the proposed design principle offers insights beyond numerical performance. We validate AGNI on 47 real-world datasets from the ADBench benchmark Ye et al. (2023b), where it achieves the best overall ranking among 15 classical and deep-learning baselines. Our contributions are summarized as follows:

- We introduce a dual-purpose attention mechanism that simultaneously governs feature masking and neighbor retrieval, offering a unified view of intra- and inter-instance structure.
- We propose a reconstruction-based SSL framework that leverages this dual role of attention to generate challenging and informative pretext tasks.
- We demonstrate both quantitative superiority—achieving consistent improvements across diverse datasets—and qualitative novelty, where AGNI's asymmetric neighbor retrieval naturally distinguishes normal from anomalous samples.

# 1 RELATED WORKS

## 1.1 CLASSICAL ANOMALY DETECTION

Anomaly detection in tabular data has long been studied under the one-class classification setting, where only normal instances are available during training. Traditional methods fall into several categories, including distance-based scoring, classification boundaries, and reconstruction methods (Ruff et al., 2021). Distance-based methods such as Isolation Forest (Liu et al., 2008), k-Neareset Neighbors (KNN) (Ramaswamy et al., 2000), and Local Outlier Factor (LOF) (Breunig et al., 2000) assign anomaly scores based on local proximity or density deviations. One-class SVM (Schölkopf et al., 1999) is a representative classification-based method that aims to enclose normal data within a learned boundary. Reconstruction-based methods, such as PCA (Shyu et al., 2003), project data to a lower-dimensional space and identify anomalies through reconstruction errors. Despite their simplicity and computational efficiency, these classical approaches often struggle to capture complex, nonlinear feature interactions in real-world datasets.

## 1.2 DEEP LEARNING FOR ONE-CLASS ANOMALY DETECTION

To address the shortcomings of classical methods, deep learning-based anomaly detection approaches have been proposed. Autoencoder-based models such as DAGMM (Zong et al., 2018) reconstruct normal inputs and detect samples with high reconstruction errors as anomalies. Other methods, like DeepSVDD (Ruff et al., 2018) and DROCC (Goyal et al., 2020), learn compact representations or decision surfaces that enclose normal data, excluding anomalies. Recently, self-supervised learning has been applied to one-class classification with notable success. ICL (Shenkar & Wolf, 2022) maximizes mutual information between different feature groups within each sample. MCM (Yin et al., 2024) employs learnable soft masking to learn feature interactions. They commonly rely on static transformation strategies or intra-instance objectives, lacking mechanisms for adapting tasks to instance-specific contexts or modeling relational structures across samples. In parallel, diffusion-based methods have emerged as a promising direction. DTE (Livernoche et al.,

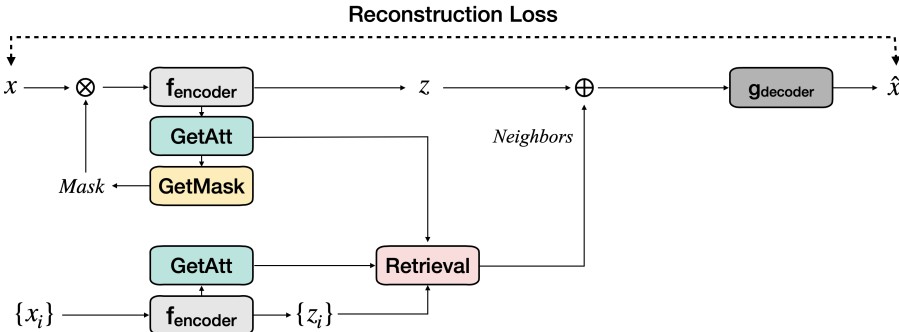

Figure 1: Overview of the proposed AGNI architecture. `GetAtt` computes attention-based feature importance scores, which are passed to `GetMask` to generate masks. The retrieval module selects neighbors with similar feature-wise importance based on these scores. Finally, the decoder reconstructs the input using the masked input and the retrieved neighbor representations, and the model is trained with a reconstruction loss.

2023) estimates the posterior distribution over diffusion time for each input sample and uses either the mode or the mean of this distribution as the anomaly score. One recent attempt to leverage inter-sample structure (Thimonier et al., 2024) incorporates retrieved neighbors into reconstruction, but its weighted aggregation strategy showed limited generalization across real-world datasets. Our work differs from prior approaches in two key aspects. First, instead of relying on static or uniform masking strategies, we use transformer attention to identify contextually important features and construct more informative reconstruction tasks. Second, unlike methods that reconstruct samples independently, we incorporate attention-wise similar neighbors during decoding, enabling relational context to enhance anomaly detection.

## 2 METHODS

We propose AGNI, a reconstruction-based SSL framework for one-class classification in tabular data (see Figure 1). AGNI consists of two key components: (1) attention-guided masking and (2) neighbor-informed reconstruction.

### 2.1 PROBLEM FORMULATION

We consider the one-class classification in tabular data, where only normal samples are available during training. Given a training set $\mathcal{D}_{\text{train}} = \{\mathbf{x}_1, \mathbf{x}_2, \ldots, \mathbf{x}_N\}$, with each $\mathbf{x}_i \in \mathbb{R}^d$ drawn from the normal data distribution $P_{\text{normal}}$, our objective is to detect unseen anomalies at test time by measuring how well a model can reconstruct the original input.

To this end, we train an encoder $f_\theta$ and a decoder $g_\phi$ to reconstruct the original input from a masked version produced by a masking operator $\mathcal{M}$:

$$\min_{\theta,\phi} \mathbb{E}_{\mathbf{x} \sim \mathcal{D}_{\text{train}}} \left[ \|\mathbf{x} - g_\phi(f_\theta(\mathcal{M}(\mathbf{x})))\|_2^2 \right] \quad (1)$$

The key challenge lies in designing $\mathcal{M}$ to construct challenging pretext tasks that capture instance-specific feature dependencies. At test time, the anomaly score is defined as the reconstruction error:

$$\text{AnomalyScore}(\mathbf{x}) := \|\mathbf{x} - g_\phi(f_\theta(\mathcal{M}(\mathbf{x})))\|_2^2 \quad (2)$$

AGNI incorporates two components to effectively model normal data structures:

- **Attention-Guided Masking**: This component masks structurally significant features identified via self-attention, compelling the model to learn inter-feature dependencies by reconstructing them from the remaining context.

- **Neighbor-Informed Reconstruction**: During decoding, the model integrates representations from attention-similar samples within the batch, providing contextual information that aids in reconstructing masked features and *implicitly* capturing inter-sample relationships.

Together, these components enable AGNI to learn both intra-instance structures and inter-instance contexts, enhancing anomaly detection performance. Detailed descriptions of the masking and retrieval components are provided in the following sections, respectively.

## 2.2 ATTENTION-GUIDED MASKING

AGNI employs a transformer-based autoencoder architecture. Given an input $\mathbf{x} \in \mathbb{R}^d$, we first embed each feature into tokens using a token embedding scheme (Gorishniy et al., 2021). The embedded input is processed through a transformer encoder with $L$ layers and $H$ attention heads to extract feature representations and attention scores. We leverage self-attention scores to identify and mask the most informative features. For each input $\mathbf{x}$, we extract attention weights from the final encoder layer and compute an importance score for each feature:

$$s_i = \texttt{GetAtt}(a_{j,i}^h) = \max_{1 \leq h \leq H} \left( \max_{1 \leq j \leq d} a_{j,i}^h \right) \tag{3}$$

where $a_{j,i}^h$ represents the attention weight from token $j$ to token $i$ in head $h$. The importance score, $s_i$ captures the maximum contextual relevance of feature $i$ across all attention heads and positions. The use of the maximum operation benefits both training and inference. During training, it encourages the model to focus on the most important feature relationships when predicting masked features, reinforcing the learning of inter-feature dependencies. At inference time, it helps amplify the effect of disrupted relationships in anomalous samples, resulting in more distinctive reconstruction errors.

The `GetMask` module selects the top-$\rho$ fraction of features based on importance scores and generates a binary mask. Let $k = \lfloor \rho \cdot d \rfloor$ be the number of features to mask. The mask $\mathbf{m} \in \{0,1\}^d$ is defined as:

$$m_i = \begin{cases} 0 & \text{if } i \in \mathcal{S}_{\text{top-}\rho} \\ 1 & \text{otherwise} \end{cases} \tag{4}$$

where $\mathcal{S}_{\text{top-}\rho}$ contains the indices of the top-$k$ features. The masked input is then computed as $\tilde{\mathbf{x}} = \mathbf{m} \odot \mathbf{x}$. By masking features with high attention scores, our method removes structurally important information from each instance. This forces the model to reconstruct these features from the remaining ones, encouraging the learning of both inter-feature dependencies and instance-specific feature relevance.

## 2.3 NEIGHBOR-INFORMED RECONSTRUCTION

To enhance reconstruction with contextual information from attention-similar samples and implicitly model inter-sample relationships, we introduce a neighbor-informed reconstruction. For each sample, we identify structurally similar neighbors by computing cosine similarity between their attention-derived importance vectors:

$$\text{sim}(\mathbf{x}_i, \mathbf{x}_j) = \frac{\mathbf{s}_i \cdot \mathbf{s}_j}{\|\mathbf{s}_i\| \cdot \|\mathbf{s}_j\|} \tag{5}$$

Our attention-based retrieval in AGNI differs from conventional embedding-based methods, which compare samples based on aggregated sample embeddings. Instead, we compare attention score distributions that reflect feature-level interaction patterns, allowing retrieval of structurally similar instances even when raw feature values differ. We find that this strategy leads to more meaningful neighbor selection, as discussed in Discussion Section.

The `Retrieval` module identifies the top-$M$ most similar neighbors and concatenates their latent representations with the representation of masked input, resulting in a combined context vector:

$$\mathbf{c}_i = [\mathbf{z}_i; \mathbf{z}_{i,1}; \ldots; \mathbf{z}_{i,M}] \tag{6}$$

In AGNI, we use concatenation to preserve individual neighbor representations, assuming that each neighbor may provide distinct contextual information identified through attention-based retrieval. In contrast, alternative fusion strategies such as averaging may dilute meaningful signals by blending diverse structural patterns across neighbors. We further evaluate these alternatives in Discussion Section.

## 2.4 Training and Inference

In AGNI, we train the model using mean squared error loss:

$$\mathcal{L}_{\text{recon}} = \frac{1}{B} \sum_{i=1}^{B} \|\mathbf{x}_i - \hat{\mathbf{x}}_i\|_2^2, \quad \text{where } \hat{\mathbf{x}}_i = g_\phi(\mathbf{c}_i) \tag{7}$$

where $B$ is the batch size. During training, neighbors are retrieved from within each batch, implicitly encouraging the model to learn relationships across the normal data distribution.

During inference, test samples are processed in batches where neighbors are selected based on attention similarity. The anomaly score is defined as:

$$\text{AnomalyScore}(\mathbf{x}_{\text{test}}) = \|\mathbf{x}_{\text{test}} - \hat{\mathbf{x}}_{\text{test}}\|_2^2 \tag{8}$$

with higher values indicating greater deviation from learned normal patterns.

## 3 Experiments

### 3.1 Experimental Settings

**Evaluation Benchmark**   To evaluate the effectiveness of our proposed AGNI, we utilize AD-Bench Han et al. (2022), a comprehensive benchmark specifically designed for tabular anomaly detection. We conduct extensive experiments on all 47 real-world tabular datasets provided in AD-Bench, spanning domains including healthcare, finance, cybersecurity, and others such as biology and industrial monitoring. These datasets inherently contain natural noise observed in real-world data collection and operation, thereby providing a realistic evaluation environment. The consistent performance of AGNI across such noisy and diverse datasets demonstrates its robustness and practical applicability. The statistical details of each dataset are provided in Appendix.

**Baseline Methods**   We conducted extensive experiments by comparing our method against 15 representative baselines from both classical and deep learning-based anomaly detection methods. The classical methods include Isolation Forest (IForest) (Liu et al., 2008), k-Nearest Neighbors (kNN) (Ramaswamy et al., 2000), Local Outlier Factor (LOF) (Breunig et al., 2000), One-Class SVM (OC-SVM) (Tax & Duin, 2004), and Principal Component Analysis (PCA) (Shyu et al., 2003). The deep learning-based baselines consist of DAGMM (Zong et al., 2018), DeepSVDD (Ruff et al., 2018), DROCC (Goyal et al., 2020), GOAD (Bergman & Hoshen, 2020), and Variational Autoencoder (VAE) (Kingma et al., 2013), as well as recent advanced methods such as ICL (Shenkar & Wolf, 2022), SLAD (Xu et al., 2023), DTE (Livernoche et al., 2023), MCM (Yin et al., 2024), and DRL (Ye et al., 2025). All experiments were repeated five times with different random seeds, and the average results are reported.

**Implementation details of AGNI**   The encoder consists of a transformer backbone followed by a three-layer multilayer perceptron (MLP) with ReLU activations. The decoder was also implemented as a three-layer MLP with ReLU activations. We applied a token embedding scheme to input the features into the transformer model (Gorishniy et al., 2021). By default, the number of tokens was set to 24, but for certain datasets, the number of heads and tokens were reduced for computational efficiency. The network parameters were optimized using the AdamW optimizer with a learning rate $1 \times 10^{-4}$ and a weight decay of $1 \times 10^{-5}$. All experiments were conducted on a single NVIDIA GeForce RTX 3090 GPU. Further implementation details are provided in Appendix.

**Evaluation**   Following previous studies (Yin et al., 2024; Ye et al., 2025; Shenkar & Wolf, 2022), we used 50% of the normal samples for training, and the remaining normal samples along with all abnormal samples as test data. The number of samples was limited to 50,000 per dataset (Livernoche et al., 2023). As evaluation criteria, we adopted widely used metrics in anomaly detection: Area Under the Receiver Operating Characteristic Curve (AUC-ROC), Area Under the Precision-Recall Curve (AUC-PR), and F1 score.

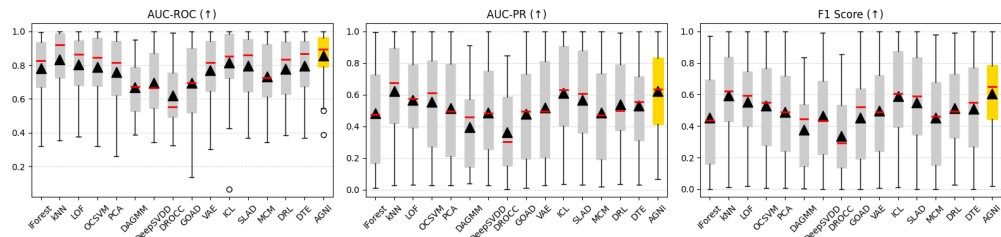

(a) Box plots comparing the average performance (AUC-ROC, AUC-PR, and F1 score) of 16 algorithms across ADBench benchmark datasets. Higher values indicate better performance. Black triangles represent the mean values.

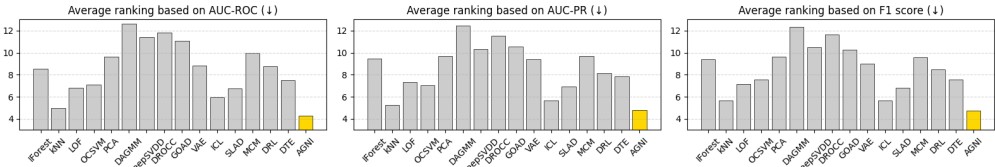

(b) Average rankings of all methods based on AUC-ROC, AUC-PR, and F1 score across ADBench benchmark datasets. Lower values indicate better performance.

Figure 2: Comparison of average performance and rankings across ADBench benchmark datasets. Subfigure (a) shows the distribution of AUC-ROC, AUC-PR, and F1 scores, while (b) reports the corresponding average rankings of each method.

Table 1: AUC-ROC performance comparison under varying anomaly ratios. Datasets are grouped into three categories based on their anomaly proportions: low (< 3.2%), medium (3.2–10.2%), and high (>10.2%). AGNI consistently achieves the highest performance, demonstrating robustness to anomaly ratio shifts.

| Ratio | IForest | kNN | LOF | OC SVM | PCA | DA GMM | Deep SVDD | DR OCC | GOAD | VAE | ICL | SLAD | MCM | DRL | DTE | **AGNI** |
|-------|---------|-----|-----|--------|-----|--------|-----------|--------|------|-----|-----|------|-----|-----|-----|----------|
| Low | 0.832 | 0.871 | 0.851 | 0.829 | 0.819 | 0.710 | 0.703 | 0.683 | 0.706 | 0.819 | 0.841 | 0.799 | 0.770 | 0.835 | 0.856 | **0.908** |
| Mid | 0.809 | 0.872 | 0.853 | 0.819 | 0.789 | 0.681 | 0.704 | 0.601 | 0.719 | 0.796 | 0.873 | 0.834 | 0.765 | 0.810 | 0.824 | **0.878** |
| High | 0.707 | 0.762 | 0.708 | 0.719 | 0.676 | 0.621 | 0.683 | 0.575 | 0.664 | 0.694 | 0.741 | 0.757 | 0.666 | 0.695 | 0.713 | **0.790** |

## 3.2 ANOMALY DETECTION RESULTS

We evaluate all methods on 47 tabular benchmark datasets from ADBench. Figure 2(a) shows AGNI's strong and stable performance, with competitive mean and median AUC-ROC and low variance. Its consistent top ranking across metrics is further confirmed in Figure 2(b). Notably, AGNI outperforms MCM (Yin et al., 2024), a recent soft masking method, confirming the benefit of our attention-guided design. The complete experimental results for all baselines and AGNI in terms of AUC-ROC, AUC-PR, and F1 score are provided in the Appendix. As shown in Tables A2, A3, and A4, AGNI consistently demonstrates outstanding performance across all metrics.

## 3.3 ROBUSTNESS TO ANOMALY RATIO

Our method targets the OCC setting, where training relies solely on normal samples and is thus unaffected by anomaly proportions. At test time, although anomaly ratios may influence evaluation metrics (e.g., AUC-ROC), AGNI's reconstruction-based detection remains stable across different proportions. To empirically validate this robustness, we partition the ADBench benchmark into three groups according to their anomaly proportions: low (< 3.2%), medium (3.2–10.2%), and high (> 10.2%). Each group contains a comparable number of datasets, enabling a balanced evaluation.

Table 1 reports the mean of AUC-ROC for AGNI and 15 baselines across the three anomaly-ratio categories. The results reveal that AGNI consistently achieves the highest mean performance in all categories, while maintaining the lowest variance, confirming its robustness to varying anomaly

proportions. These findings strengthen the evidence that AGNI not only improves average detection accuracy but also delivers reliable performance across different data imbalance conditions.

## 4 ANALYSIS

### 4.1 ABLATION STUDY

We performed an ablation study to evaluate the effectiveness of AGNI's attention-guided masking strategy and neighbor-informed reconstruction. To conduct the ablation study, we constructed five subsets from diverse domains, including healthcare, document analysis, linguistics, and biology (*annthyroid*, *cardiotocography*, *stamps*, *vowels*, and *yeast*). In the following experiments, most of the reported results are based on the average performance over predefined subsets. Detailed results for each individual dataset are provided in Appendix.

**Impact of Varying the Number of Retrieved Neighbors** We investigate how the number of retrieved neighbors ($M$) affects performance by varying $M$ from 1 to 3 while keeping the masking ratio fixed. Figure 3 presents results on two representative datasets. Performance consistently improves with more neighbors across all evaluation metrics, supporting our hypothesis that incorporating multiple similar neighbors provides more informative context for reconstructing masked features while strengthening the model's understanding of inter-sample relationships. However, the gains begin to plateau at $M = 3$, indicating a trade-off between contextual diversity and redundancy. Based on this observation, we set $M = 3$ as the default in all experiments.

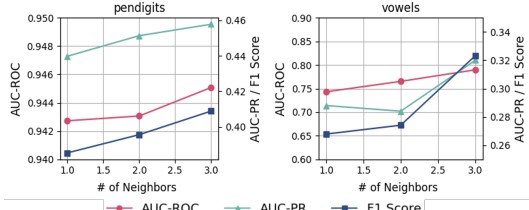

Figure 3: Effect of the number of retrieved neighbors on anomaly detection performance. We fix the augmentation ratio and vary the number of retrieved neighbors.

**Effect of batch size** We investigate how the test-time batch size affects anomaly detection performance, since AGNI retrieves neighbors within each batch. Larger batch size provides a broader pool of candidates for identifying attention score similar neighbors, enabling AGNI to better capture inter-sample relationships and learn the underlying structure of the normal data distribution. Table 2 shows AUC-ROC scores for batch sizes ranging from 4 to 256. Performance improves steadily as batch size increases from 4 to 64, with average AUC-ROC rising from 0.768 to 0.800. Beyond 64, performance plateaus or even slightly degrades. Our dataset-specific batch size assignment strategy are all detailed in the Appendix.

Table 2: AUC-ROC across batch sizes

| Batch size | 4 | 8 | 16 | 64 | 128 | 256 |
|---|---|---|---|---|---|---|
| annthyroid | 0.921 | 0.939 | 0.947 | 0.955 | 0.956 | **0.958** |
| cardiotocography | 0.762 | 0.774 | 0.785 | 0.793 | **0.794** | **0.794** |
| stamps | 0.915 | 0.935 | **0.938** | 0.932 | 0.934 | 0.934 |
| vowels | 0.723 | 0.752 | 0.762 | **0.790** | 0.778 | 0.773 |
| yeast | 0.518 | 0.520 | 0.514 | **0.529** | 0.528 | 0.519 |
| Average | 0.768 | 0.784 | 0.789 | **0.800** | 0.798 | 0.796 |

**Component-wise analysis** We conduct an ablation study to evaluate the contribution of each component in AGNI, as shown in Table 3. Starting from a vanilla transformer based autoencoder baseline without either proposed component (Setting A), we incrementally add attention-guided masking component and neighbor-informed reconstruction component. Adding attention-guided masking alone (A → B) improves the F1 score, indicating that masking contextually important features enhances discriminative ability. Incorporating neighbor-informed reconstruction component instead (A → C) yields a gain

Table 3: Ablation study on the effect of each proposed component.

| Setting | Mask | Neighbor | AUC-ROC | PR | F1 | Time (s) |
|---|---|---|---|---|---|---|
| A | ✗ | ✗ | 0.768 | 0.555 | 0.538 | 0.022 |
| B | ✓ | ✗ | 0.764 | 0.559 | 0.560 | 0.022 |
| C | ✗ | ✓ | 0.790 | 0.578 | 0.555 | 0.031 |
| D (AGNI) | ✓ | ✓ | **0.808** | **0.614** | **0.597** | 0.032 |

Table 4: Evaluation results for configuration-wise variants

| Variant | Masking | Neighbor selection criteria | Fusion strategy | AUC-ROC | AUC-PR | F1 score |
|---|---|---|---|---|---|---|
| 1 | Attention-guided | Attention score | $\left[\frac{1}{M+1}\left(z + \sum_i^M \mathbf{z}_i\right)\right]$ | 0.768 | 0.509 | 0.487 |
| 2 | Attention-guided | Attention score | $[z; \frac{1}{M}\sum_i^M \mathbf{z}_i]$ | 0.791 | 0.578 | 0.566 |
| 3 | Attention-guided | Embedding sim. | $[z; z_1; \ldots; z_M]$ | 0.801 | 0.575 | 0.550 |
| 4 | Random | Attention score | $[z; z_1; \ldots; z_M]$ | 0.712 | 0.483 | 0.501 |
| 5 (AGNI) | Attention-guided | Attention score | $[z; z_1; \ldots; z_M]$ | **0.808** | **0.614** | **0.597** |

in AUC-ROC, suggesting its effectiveness in both structural patterns and inter-sample relationships within the normal data distribution. The full model combining both components (AGNI) achieves the best performance across all metrics. These improvements demonstrate that the two components provide complementary benefits: attention-guided masking creates informative learning objectives, while neighbor-informed reconstruction leverages relationships between similar samples to better characterize the normal data manifold. We further measured the inference time per batch. Incorporating attention-guided masking resulted in negligible change, whereas adding neighbor-informed reconstruction introduced a delay of approximately 9 ms. Nonetheless, this delay is minimal relative to the performance gains it brings and thus does not hinder practical applicability.

## 5 DISCUSSION

### 5.1 IMPACT OF CONFIGURATION VARIANTS

The proposed attention-guided masking component and neighbor-informed reconstruction component in AGNI involve several configuration-level alternatives. This section examines the performance impact of different strategies for each configuration. The results are averaged over the subset of five datasets introduced in Analysis Section. Table 4 summarizes the evaluated alternatives and their corresponding results.

**Input–Neighbor Fusion Variants**  Given our neighbor-informed reconstruction component, we investigated how to effectively utilize the selected neighbors as context for reconstruction. We evaluated three fusion strategies that differ in how the input representation is combined with the retrieved neighbor representation, as shown in variants 1, 2, and 5 of Table 4. Proposed AGNI (variant 5), which concatenates the input with each neighbor embedding individually, achieves superior performance compared to two alternatives: variant 1, which averages the input and neighbor representations, and variant 2, which concatenates the input representation with the averaged neighbor representation. Averaging the selected neighbors may blur their unique contributions, thereby undermining the precision of attention-based retrieval. The superior performance of variant 5 suggests that preserving individual neighbor representations enables the decoder to more effectively leverage the contextual information captured by the attention mechanism.

**Neighbor selection: attention-based vs embedding-based**  We compare AGNI's attention-based neighbor retrieval with an alternative based on latent embeddings. In the embedding-based variant, neighbors are selected by computing cosine similarity between max-pooled encoder outputs. We chose max-pooling as it preserves the most salient features from each dimension, though other pooling strategies such as mean-pooling or attention-weighted pooling could also be considered. However, these methods operate on aggregated representations and could not directly reflect feature-wise importance patterns. In contrast, AGNI uses similarity over attention scores where each element represents the maximum attention a feature receives across all heads and positions. This preserves feature-level relationships that are lost in embedding-based approaches. As shown in variants 3, 5 of Table 4, attention-based retrieval achieves consistently higher performance across all evaluation metrics. This suggests that attention-based similarity leads to more relevant neighbor selection in our anomaly detection framework. Exploring alternative pooling strategies and distance metrics for embedding-based retrieval remains an interesting direction for future work.

**Comparison of masking strategies**  We compare the attention-guided masking component proposed in AGNI with a baseline that masks randomly selected features. For each dataset, both meth-

ods are evaluated using their respective masking ratios. As shown in variants 4 and 5 of Table 4, attention-guided masking consistently outperforms random masking across all metrics (AUC-ROC, AUC-PR, and F1 score). This improvement highlights the importance of using attention scores to target structurally salient features rather than perturbing inputs arbitrarily. By masking features that the model itself identifies as most informative, attention-guided masking generates more challenging and context-aware reconstruction tasks, which in turn encourage the model to capture fine-grained feature dependencies. Random masking, by contrast, often removes irrelevant or redundant attributes, resulting in weaker supervision and limited gains. These results confirm that attention-guided masking provides a principled mechanism for constructing effective self-supervised signals in tabular anomaly detection.

## 5.2 QUALITATIVE ANALYSIS OF NEIGHBOR SELECTION

To validate our attention-based neighbor retrieval mechanism, we visualize the selected neighbors using t-SNE of the input space. As shown in Figure 4, neighbors of normal samples (blue triangles) cluster tightly around their inputs, while neighbors of anomalous samples (yellow stars) are widely scattered. This pattern illustrates how our method captures inter-sample relationships: normal samples successfully retrieve structurally similar neighbors, forming coherent local contexts that enable accurate reconstruction. Conversely, anomalous samples lack structurally similar neighbors in the dataset, resulting in dispersed neighbor sets that provide weak contextual support.

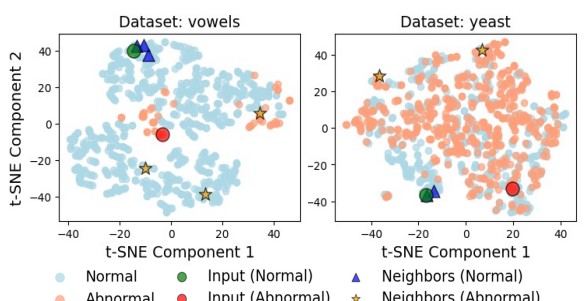

Figure 4: t-SNE visualizations of the input space for two datasets. Red and green dots represent abnormal and normal inputs. Blue triangles and yellow stars indicate retrieved neighbors. Normal inputs yield compact, aligned neighborhoods, while abnormal inputs retrieve scattered samples.

This asymmetric retrieval behavior—compact neighborhoods for normal samples versus scattered neighbors for anomalies—confirms that our attention mechanism effectively identifies structural similarities within the normal data distribution while naturally amplifying reconstruction difficulty for outliers.

## 6 CONCLUSION

In this work, we introduced AGNI, a self-supervised framework for one-class classification in tabular anomaly detection that unifies intra-instance and inter-sample modeling through a dual supervisory use of attention. By employing attention-guided masking to enforce fine-grained feature dependencies and neighbor-informed reconstruction to exploit structural similarities across samples, AGNI transforms attention from a representational mechanism into a coordinating learning signal. Extensive evaluations on 47 real-world datasets demonstrate that AGNI achieves consistent improvements over 15 classical and deep-learning baselines, obtaining the best average rank across AUC-ROC, AUC-PR, and F1 score, and delivering the best overall performance. Beyond numerical gains, AGNI reveals interpretable qualitative behavior: normal samples form compact neighborhoods while anomalies yield scattered ones, offering intuitive evidence of its structural novelty. Taken together, these results highlight AGNI's effectiveness in detecting subtle, context-dependent anomalies and underscore the promise of designing supervisory signals that jointly capture instance-specific and population-level structure. We believe this design principle opens new directions for future research, including extensions to semi-supervised or multi-modal anomaly detection, integration with generative modeling, and the exploration of attention-driven supervisory signals in broader tabular learning tasks.

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

# Attention-Guided Masking and Neighbor-Informed Reconstruction for Tabular Anomaly Detection

We used a large language model (LLM) only to polish grammar and improve readability of the manuscript. All scientific content, including methods, experiments, and analyses, was entirely conceived and produced by the authors.

## A    DATASETS DETAILS

Table A1: Dataset properties.

| Dataset | Samples | Features | Anomalies | Anomaly(%) | Category |
|---|---|---|---|---|---|
| aloi | 49534 | 27 | 1508 | 3.04 | Image |
| annthyroid | 7200 | 6 | 534 | 7.42 | Healthcare |
| backdoor | 95329 | 196 | 2329 | 2.44 | Network |
| breastw | 683 | 9 | 239 | 34.99 | Healthcare |
| campaign | 41188 | 62 | 4640 | 11.27 | Finance |
| cardio | 1831 | 21 | 176 | 9.61 | Healthcare |
| cardiotocography | 2114 | 21 | 466 | 22.04 | Healthcare |
| celeba | 202599 | 39 | 4547 | 2.24 | Image |
| census | 299285 | 500 | 18568 | 6.2 | Sociology |
| cover | 286048 | 10 | 2747 | 0.96 | Botany |
| donors | 619326 | 10 | 36710 | 5.93 | Sociology |
| fault | 1941 | 27 | 673 | 34.67 | Physical |
| fraud | 284807 | 29 | 492 | 0.17 | Finance |
| glass | 214 | 7 | 9 | 4.21 | Forensic |
| hepatitis | 80 | 19 | 13 | 16.25 | Healthcare |
| http | 567498 | 3 | 2211 | 0.39 | Web |
| internetads | 1966 | 1555 | 368 | 18.72 | Image |
| ionosphere | 351 | 32 | 126 | 35.9 | Oryctognosy |
| landsat | 6435 | 36 | 1333 | 20.71 | Astronautics |
| letter | 1600 | 32 | 100 | 6.25 | Image |
| lymphography | 148 | 18 | 6 | 4.05 | Healthcare |
| magic | 19020 | 10 | 6688 | 35.16 | Physical |
| mammography | 11183 | 6 | 260 | 2.32 | Healthcare |
| mnist | 7603 | 100 | 700 | 9.21 | Image |
| musk | 3062 | 166 | 97 | 3.17 | Chemistry |
| optdigits | 5216 | 64 | 150 | 2.88 | Image |
| pageBlocks | 5393 | 10 | 510 | 9.46 | Document |
| pendigits | 6870 | 16 | 156 | 2.27 | Image |
| pima | 768 | 8 | 268 | 34.9 | Healthcare |
| satellite | 6435 | 36 | 2036 | 31.64 | Astronautics |
| satimage-2 | 5803 | 36 | 71 | 1.22 | Astronautics |
| shuttle | 49097 | 9 | 3511 | 7.15 | Astronautics |
| skin | 245057 | 3 | 50859 | 20.75 | Image |
| smtp | 95156 | 3 | 30 | 0.03 | Web |
| spambase | 4207 | 57 | 1679 | 39.91 | Document |
| speech | 3686 | 400 | 61 | 1.65 | Linguistics |
| stamps | 340 | 9 | 31 | 9.12 | Document |
| thyroid | 3772 | 6 | 93 | 2.47 | Healthcare |
| vertebral | 240 | 6 | 30 | 12.5 | Biology |
| vowels | 1456 | 12 | 50 | 3.43 | Linguistics |
| waveform | 3443 | 21 | 100 | 2.9 | Physics |
| wbc | 223 | 9 | 10 | 4.48 | Healthcare |
| wdbc | 367 | 30 | 10 | 2.72 | Healthcare |
| wilt | 4819 | 5 | 257 | 5.33 | Botany |
| wine | 129 | 13 | 10 | 7.75 | Chemistry |
| wpbc | 198 | 33 | 47 | 23.74 | Healthcare |
| yeast | 1484 | 8 | 507 | 34.16 | Biology |

This study utilized 47 real-world tabular anomaly detection datasets provided by the ADBench benchmark suite Han et al. (2022). These datasets span a broad spectrum of application domains in which anomalies are known to arise in practice, including healthcare, finance, network security, web activity, biology, imaging, and astronautics. Table A1 presents the detailed properties of the datasets used in our experiments. Moreover, the benchmark covers a wide range of data characteristics, including sample size (from under 100 to over 600,000 instances), feature dimensionality (from as few as 3 to over 1,500), and anomaly ratios (ranging from 0.03% to nearly 40%). Such diversity ensures a comprehensive evaluation, capturing the complex and varied nature of real-world anomaly detection scenarios.

## B  IMPLEMENTATION DETAILS

The encoder used in AGNI consists of a transformer followed by a three-layer multilayer perceptron (MLP) with ReLU activations. All features are treated as continuous variables and embedded using a token embedding scheme before being passed into the transformer Gorishniy et al. (2021). The transformer comprises six layers with eight attention heads and a token dimension of 24. For the *census* and *internetads* datasets, we reduce the number of attention heads and the token dimension to four and eight, respectively, for computational efficiency. The decoder shares the same three-layer MLP architecture used in the encoder. We train the network using the AdamW optimizer with a learning rate of $1 \times 10^{-4}$ and a weight decay of $1 \times 10^{-5}$. All experiments are conducted on a single NVIDIA GeForce RTX 3090 GPU.

The batch size was determined separately for the training and test sets based on the number of samples in each. Specifically, we used a batch size of 64 for datasets with fewer than 1,000 samples; 128 for datasets with 1,000 to 4,999 samples; 256 for datasets with 5,000 to 9,999 samples; 512 for datasets with 10,000 to 49,999 samples; and 1,024 for datasets with 50,000 or more samples.

## C  COMPREHENSIVE EVALUATION RESULTS

This section presents the comprehensive evaluation results that supplement the main text. First, we provide the AUC-PR and F1 scores of AGNI and 15 baseline methods evaluated across 47 real-world tabular anomaly detection datasets Han et al. (2022). Second, we include the full component-wise analysis results based on the five-dataset subset used in the main text ablation analysis and discussion sections. We also present the complete results corresponding to the study on the impact of configuration variant discussed in the main text. Additionally, we examine performance across datasets categorized by anomaly ratios to assess the method's robustness under different anomaly distributions. Finally, we report the optimal masking ratios and the number of neighbors selected for each dataset.

**AUC-ROC scores of AGNI on full benchmark datasets**    Based on the AUC-ROC (↑) scores summarized in Table A2, AGNI demonstrates strong performance across 47 tabular anomaly detection datasets. It achieves the best average scores and the best average rank across datasets.

Table A2: Comparison of AUC-ROC (↑) scores across 47 tabular anomaly detection datasets. Each value represents the average over five independent runs with different random seeds. For each dataset, the best result is highlighted in **bold**, and the second-best is underlined.

| Dataset | IForest | kNN | LOF | OC-SVM | PCA | DA-GMM | Deep-SVDD | DRO-CC | GOAD | VAE | ICL | SLAD | MCM | DRL | DTE | AGNI |
|---|---|---|---|---|---|---|---|---|---|---|---|---|---|---|---|---|
| aloi | 0.507 | 0.510 | 0.488 | 0.543 | 0.540 | 0.508 | 0.509 | 0.500 | 0.480 | 0.540 | 0.475 | 0.508 | 0.543 | 0.518 | 0.505 | **0.547** |
| annthyroid | 0.903 | 0.928 | 0.886 | 0.884 | 0.852 | 0.722 | 0.550 | 0.889 | 0.810 | 0.854 | 0.811 | 0.933 | 0.707 | 0.583 | **0.975** | 0.956 |
| backdoor | 0.749 | 0.938 | **0.953** | 0.625 | 0.646 | 0.544 | 0.911 | 0.943 | 0.529 | 0.647 | 0.936 | 0.500 | 0.924 | 0.930 | 0.904 | 0.934 |
| breastw | 0.995 | 0.991 | 0.889 | 0.994 | 0.992 | 0.895 | 0.970 | 0.473 | 0.989 | 0.992 | 0.983 | 0.995 | **0.998** | 0.985 | 0.939 | 0.994 |
| campaign | 0.736 | 0.785 | 0.706 | 0.777 | 0.771 | 0.615 | 0.622 | 0.500 | 0.479 | 0.771 | 0.809 | 0.767 | 0.697 | 0.725 | 0.790 | **0.852** |
| cardio | 0.933 | 0.920 | 0.922 | 0.956 | 0.965 | 0.779 | 0.654 | 0.621 | 0.960 | 0.966 | 0.800 | 0.831 | 0.806 | 0.879 | 0.878 | **0.979** |
| cardiotocography | 0.742 | 0.621 | 0.645 | 0.752 | 0.789 | 0.671 | 0.477 | 0.460 | 0.761 | 0.789 | 0.542 | 0.473 | **0.843** | 0.762 | 0.619 | 0.794 |
| celeba | 0.712 | 0.731 | 0.437 | 0.798 | 0.805 | 0.638 | 0.562 | 0.689 | 0.438 | 0.803 | 0.722 | 0.674 | 0.734 | 0.758 | **0.826** | 0.816 |
| census | 0.625 | **0.723** | 0.585 | 0.700 | 0.705 | 0.522 | 0.542 | 0.554 | 0.352 | 0.705 | 0.706 | 0.579 | 0.660 | 0.575 | 0.691 | 0.721 |
| cover | 0.863 | 0.975 | 0.992 | 0.962 | 0.944 | 0.759 | 0.491 | 0.958 | 0.138 | 0.944 | 0.893 | 0.740 | 0.484 | 0.972 | 0.955 | **0.993** |
| donors | 0.894 | 0.995 | 0.970 | 0.921 | 0.881 | 0.622 | 0.730 | 0.742 | 0.336 | 0.886 | **0.999** | 0.885 | 0.905 | 0.979 | 0.979 | 0.885 |
| fault | 0.559 | 0.587 | 0.474 | 0.572 | 0.559 | 0.528 | 0.543 | 0.557 | 0.589 | 0.559 | 0.606 | **0.639** | 0.590 | 0.603 | 0.588 | 0.620 |
| fraud | 0.947 | 0.954 | 0.944 | 0.956 | 0.954 | 0.853 | 0.831 | 0.500 | 0.698 | 0.955 | 0.928 | 0.946 | 0.932 | 0.942 | 0.888 | **0.963** |
| glass | 0.811 | 0.920 | 0.888 | 0.697 | 0.734 | 0.653 | 0.837 | 0.649 | 0.590 | 0.726 | **0.994** | 0.860 | 0.694 | 0.759 | 0.781 | 0.877 |
| hepatitis | 0.827 | 0.965 | 0.669 | 0.906 | 0.845 | 0.702 | 0.996 | 0.518 | 0.845 | 0.848 | **0.999** | 0.999 | 0.518 | 0.603 | 0.791 | 0.816 |
| http | 0.994 | **1.000** | 1.000 | **1.000** | 1.000 | 0.918 | 0.613 | 0.500 | 0.997 | 0.999 | 0.982 | 0.999 | 0.998 | 0.998 | 0.990 | **1.000** |
| internetads | 0.479 | 0.681 | 0.717 | 0.656 | 0.651 | 0.495 | 0.730 | 0.534 | 0.656 | 0.651 | 0.684 | 0.633 | 0.759 | 0.684 | **0.771** | 0.662 |
| ionosphere | 0.912 | 0.974 | 0.943 | 0.963 | 0.891 | 0.740 | 0.972 | 0.611 | 0.915 | 0.898 | **0.990** | 0.982 | 0.635 | 0.954 | 0.942 | 0.971 |
| landsat | 0.588 | 0.682 | 0.666 | 0.480 | 0.439 | 0.563 | 0.594 | 0.539 | 0.405 | 0.542 | 0.651 | 0.650 | 0.477 | **0.699** | 0.521 | 0.615 |
| letter | 0.320 | 0.354 | 0.448 | 0.322 | 0.303 | 0.390 | 0.364 | **0.553** | 0.311 | 0.302 | 0.427 | 0.368 | 0.344 | 0.402 | 0.371 | 0.389 |
| lymphography | 0.995 | 0.999 | 0.982 | **1.000** | 0.999 | 0.949 | 0.997 | 0.324 | 0.999 | 0.999 | **1.000** | **1.000** | 0.922 | 0.936 | **1.000** | **1.000** |
| magic.gamma | 0.771 | 0.833 | 0.834 | 0.743 | 0.706 | 0.592 | 0.630 | 0.788 | 0.695 | 0.706 | 0.756 | 0.720 | 0.831 | 0.815 | **0.873** | 0.763 |
| mammography | 0.880 | 0.876 | 0.855 | 0.886 | **0.899** | 0.760 | 0.715 | 0.818 | 0.699 | 0.896 | 0.719 | 0.745 | 0.846 | 0.887 | 0.867 | 0.896 |
| mnist | 0.866 | 0.939 | 0.929 | 0.906 | 0.902 | 0.722 | 0.664 | 0.831 | 0.901 | 0.902 | 0.901 | 0.902 | 0.940 | **0.965** | 0.895 | 0.948 |
| musk | 0.906 | **1.000** | 1.000 | **1.000** | 1.000 | 0.950 | 1.000 | 0.330 | **1.000** | 1.000 | 0.994 | **1.000** | 0.962 | 0.885 | **1.000** | 0.999 |
| optdigits | 0.811 | 0.937 | 0.967 | 0.634 | 0.582 | 0.400 | 0.395 | 0.853 | 0.675 | 0.582 | **0.972** | 0.953 | 0.668 | 0.847 | 0.851 | 0.887 |
| pageblocks | 0.826 | 0.896 | 0.913 | 0.886 | 0.861 | 0.828 | 0.784 | **0.923** | 0.880 | 0.862 | 0.884 | 0.879 | 0.724 | 0.756 | 0.900 | 0.908 |
| pendigits | 0.972 | **0.999** | 0.991 | 0.964 | 0.944 | 0.565 | 0.463 | 0.759 | 0.900 | 0.945 | 0.967 | 0.946 | 0.653 | 0.847 | 0.981 | 0.945 |
| pima | 0.743 | 0.769 | 0.705 | 0.715 | 0.723 | 0.545 | 0.580 | 0.475 | 0.623 | 0.732 | **0.797** | 0.606 | 0.719 | 0.622 | 0.653 | 0.723 |
| satellite | 0.775 | 0.822 | 0.803 | 0.739 | 0.666 | 0.728 | 0.762 | 0.734 | 0.688 | 0.741 | 0.852 | **0.875** | 0.687 | 0.747 | 0.786 | 0.790 |
| satimage-2 | 0.991 | 0.997 | 0.994 | 0.996 | 0.982 | 0.918 | 0.929 | 0.992 | 0.990 | 0.990 | 0.995 | **0.998** | 0.997 | 0.996 | 0.994 | 0.997 |
| shuttle | 0.997 | 0.999 | **1.000** | 0.996 | 0.994 | 0.846 | 0.998 | 0.500 | 0.704 | 0.994 | 0.999 | 0.999 | 0.997 | 0.996 | 0.998 | 0.999 |
| skin | 0.894 | **0.995** | 0.863 | 0.902 | 0.597 | 0.679 | 0.600 | 0.895 | 0.649 | 0.660 | 0.066 | 0.910 | 0.889 | 0.944 | 0.900 | 0.856 |
| smtp | 0.904 | 0.924 | 0.934 | 0.847 | 0.818 | 0.871 | 0.852 | 0.571 | 0.788 | 0.819 | 0.744 | 0.921 | 0.879 | 0.866 | 0.945 | **0.960** |
| spambase | **0.852** | 0.834 | 0.732 | 0.817 | 0.814 | 0.694 | 0.702 | 0.754 | 0.818 | 0.814 | 0.835 | 0.849 | 0.801 | 0.725 | 0.831 | 0.837 |
| speech | 0.377 | 0.364 | 0.375 | 0.366 | 0.364 | 0.507 | 0.489 | 0.490 | 0.366 | 0.364 | 0.489 | 0.414 | 0.421 | 0.463 | 0.386 | **0.925** |
| stamps | 0.935 | 0.959 | 0.937 | 0.937 | 0.927 | 0.801 | 0.711 | 0.501 | 0.815 | 0.933 | **0.967** | 0.820 | 0.858 | 0.855 | 0.908 | 0.932 |
| thyroid | **0.990** | 0.987 | 0.927 | 0.986 | 0.986 | 0.911 | 0.888 | 0.950 | 0.952 | 0.986 | 0.954 | 0.953 | 0.933 | 0.870 | 0.989 | 0.983 |
| vertebral | 0.456 | 0.577 | 0.643 | 0.505 | 0.421 | 0.506 | 0.448 | 0.438 | 0.467 | 0.426 | 0.792 | 0.450 | 0.341 | 0.386 | 0.418 | **0.878** |
| vowels | 0.618 | 0.822 | 0.863 | 0.759 | 0.523 | 0.425 | 0.557 | 0.547 | 0.685 | 0.521 | 0.851 | 0.850 | 0.658 | 0.836 | **0.869** | 0.790 |
| waveform | 0.723 | 0.752 | **0.760** | 0.704 | 0.647 | 0.519 | 0.599 | 0.677 | 0.650 | 0.648 | 0.687 | 0.489 | 0.394 | 0.605 | 0.650 | 0.690 |
| wbc | 0.994 | 0.991 | 0.805 | 0.996 | 0.994 | 0.868 | 0.914 | 0.442 | 0.991 | 0.993 | 0.997 | **0.998** | 0.930 | 0.905 | 0.872 | 0.945 |
| wdbc | 0.987 | 0.991 | 0.996 | 0.993 | 0.991 | 0.738 | 0.993 | 0.401 | 0.990 | 0.991 | **0.998** | 0.995 | 0.948 | 0.977 | 0.962 | 0.994 |
| wilt | 0.480 | 0.637 | 0.688 | 0.348 | 0.261 | 0.418 | 0.344 | 0.495 | 0.514 | 0.354 | 0.764 | 0.618 | 0.397 | 0.760 | 0.847 | **0.886** |
| wine | 0.939 | 0.992 | 0.984 | 0.978 | 0.938 | 0.662 | 0.922 | 0.438 | 0.941 | 0.943 | 0.999 | **1.000** | 0.941 | 0.958 | 0.402 | 0.951 |
| wpbc | 0.563 | 0.637 | 0.574 | 0.534 | 0.525 | 0.470 | 0.827 | 0.438 | 0.514 | 0.544 | **0.966** | 0.955 | 0.517 | 0.430 | 0.518 | 0.937 |
| yeast | 0.418 | 0.447 | 0.458 | 0.448 | 0.432 | 0.510 | 0.476 | 0.484 | 0.525 | 0.424 | 0.490 | 0.487 | 0.433 | 0.487 | 0.472 | **0.529** |
| Average Value | 0.782 | 0.834 | 0.803 | 0.788 | 0.761 | 0.670 | 0.697 | 0.620 | 0.696 | 0.769 | 0.817 | 0.796 | 0.733 | 0.779 | 0.797 | **0.858** |
| Average Ranking | 8.53 | 4.97 | 6.84 | 7.12 | 9.64 | 12.62 | 11.40 | 11.80 | 11.05 | 8.85 | 5.94 | 6.76 | 9.97 | 8.77 | 7.48 | **4.28** |

**AUC-PR scores of AGNI on full benchmark datasets** Based on the AUC-PR (↑) scores summarized in Table A3, AGNI demonstrates strong performance across 47 tabular anomaly detection datasets. It achieves the highest average scores and the best average rank, demonstrating consistent effectiveness across diverse domains and data characteristics.

Table A3: Comparison of AUC-PR (↑) scores across 47 tabular anomaly detection datasets. Each value represents the average over five independent runs with different random seeds. For each dataset, the best result is highlighted in **bold**, and the second-best is underlined.

| Dataset | IForest | kNN | LOF | OC-SVM | PCA | DA-GMM | Deep-SVDD | DRO-CC | GOAD | VAE | ICL | SLAD | MCM | DRL | DTE | AGNI |
|---|---|---|---|---|---|---|---|---|---|---|---|---|---|---|---|---|
| aloi | 0.058 | 0.060 | 0.065 | 0.065 | 0.065 | 0.061 | 0.062 | 0.059 | 0.057 | 0.065 | 0.055 | 0.060 | 0.085 | **0.093** | 0.058 | 0.067 |
| annthyroid | 0.590 | 0.681 | 0.535 | 0.601 | 0.566 | 0.480 | 0.278 | 0.637 | 0.587 | 0.567 | 0.458 | 0.706 | 0.344 | 0.200 | **0.828** | 0.764 |
| backdoor | 0.094 | 0.465 | 0.535 | 0.077 | 0.079 | 0.075 | 0.848 | 0.846 | 0.063 | 0.080 | **0.892** | 0.048 | 0.664 | 0.886 | 0.609 | 0.543 |
| breastw | 0.995 | 0.989 | 0.800 | 0.994 | 0.992 | 0.909 | 0.960 | 0.632 | 0.988 | 0.992 | 0.968 | 0.995 | **0.998** | 0.985 | 0.904 | 0.993 |
| campaign | 0.457 | 0.490 | 0.402 | 0.494 | 0.488 | 0.324 | 0.369 | 0.203 | 0.231 | 0.488 | 0.489 | 0.481 | 0.405 | 0.450 | 0.485 | **0.578** |
| cardio | 0.786 | 0.772 | 0.702 | 0.836 | 0.862 | 0.559 | 0.389 | 0.511 | 0.848 | 0.863 | 0.479 | 0.699 | **0.729** | 0.703 | 0.704 | **0.904** |
| cardiotocography | 0.628 | 0.574 | 0.573 | 0.662 | 0.697 | 0.597 | 0.458 | 0.439 | 0.675 | 0.697 | 0.487 | 0.494 | **0.729** | 0.656 | 0.546 | 0.708 |
| celeba | 0.117 | 0.119 | 0.036 | 0.203 | **0.209** | 0.090 | 0.071 | 0.076 | 0.040 | **0.209** | 0.097 | 0.093 | 0.093 | 0.113 | 0.153 | 0.182 |
| census | 0.142 | 0.217 | 0.137 | 0.203 | 0.200 | 0.132 | 0.153 | 0.142 | 0.087 | 0.198 | 0.212 | 0.150 | 0.160 | 0.127 | 0.177 | **0.221** |
| cover | 0.087 | 0.558 | **0.829** | 0.223 | 0.162 | 0.098 | 0.027 | 0.313 | 0.011 | 0.161 | 0.345 | 0.070 | 0.017 | 0.418 | 0.287 | 0.709 |
| donors | 0.405 | 0.891 | 0.634 | 0.427 | 0.352 | 0.195 | 0.427 | 0.302 | 0.090 | 0.360 | **0.984** | 0.462 | 0.667 | 0.753 | 0.709 | 0.395 |
| fault | 0.592 | 0.620 | 0.504 | 0.611 | 0.604 | 0.568 | 0.555 | 0.578 | 0.621 | 0.604 | 0.632 | **0.667** | 0.602 | 0.628 | 0.637 | 0.630 |
| fraud | 0.182 | 0.387 | 0.551 | 0.296 | 0.269 | 0.156 | 0.483 | 0.003 | 0.294 | 0.287 | 0.539 | 0.450 | 0.483 | 0.634 | **0.802** | 0.436 |
| glass | 0.214 | 0.423 | 0.381 | 0.268 | 0.210 | 0.186 | 0.523 | 0.231 | 0.183 | 0.185 | **0.523** | 0.411 | 0.197 | 0.287 | 0.256 | 0.399 |
| hepatitis | 0.554 | 0.903 | 0.437 | 0.776 | 0.648 | 0.544 | 0.987 | 0.349 | 0.658 | 0.645 | **0.998** | 0.998 | 0.350 | 0.406 | 0.580 | 0.607 |
| http | 0.534 | **1.000** | 0.971 | 0.999 | 0.917 | 0.575 | 0.361 | 0.007 | 0.684 | 0.904 | 0.708 | 0.881 | 0.923 | 0.990 | 0.417 | **1.000** |
| internetads | 0.292 | 0.492 | 0.504 | 0.481 | 0.470 | 0.318 | 0.516 | 0.431 | 0.474 | 0.470 | 0.600 | **0.605** | 0.494 | 0.439 | 0.558 | 0.478 |
| ionosphere | 0.917 | 0.980 | 0.946 | 0.975 | 0.909 | 0.775 | 0.981 | 0.717 | 0.932 | 0.914 | **0.991** | 0.986 | 0.743 | 0.965 | 0.962 | 0.979 |
| landsat | 0.473 | 0.548 | **0.614** | 0.370 | 0.327 | 0.403 | 0.494 | 0.376 | 0.312 | 0.403 | 0.531 | 0.451 | 0.367 | 0.523 | 0.374 | 0.425 |
| letter | 0.082 | 0.087 | 0.113 | 0.083 | 0.080 | 0.104 | 0.089 | **0.157** | 0.081 | 0.080 | 0.128 | 0.089 | 0.089 | 0.099 | 0.090 | 0.090 |
| lymphography | 0.944 | 0.992 | 0.842 | **1.000** | 0.985 | 0.735 | 0.968 | 0.309 | 0.988 | 0.986 | **1.000** | 0.999 | 0.417 | 0.437 | **1.000** | **1.000** |
| magic.gamma | 0.803 | 0.859 | 0.864 | 0.792 | 0.752 | 0.645 | 0.695 | 0.832 | 0.761 | 0.753 | 0.813 | 0.773 | 0.862 | 0.857 | **0.895** | 0.797 |
| mammography | 0.379 | 0.413 | 0.341 | 0.405 | 0.416 | 0.220 | 0.275 | 0.272 | 0.278 | 0.418 | 0.171 | 0.190 | 0.300 | **0.553** | 0.402 | 0.442 |
| mnist | 0.541 | 0.727 | 0.710 | 0.662 | 0.650 | 0.461 | 0.460 | 0.597 | 0.651 | 0.650 | 0.684 | 0.684 | 0.813 | **0.855** | 0.592 | 0.770 |
| musk | 0.404 | **1.000** | **1.000** | **1.000** | **1.000** | 0.706 | 0.999 | 0.157 | **1.000** | **1.000** | 0.922 | **1.000** | 0.591 | 0.382 | **1.000** | 0.996 |
| optdigits | 0.154 | 0.291 | 0.436 | 0.069 | 0.060 | 0.050 | 0.045 | 0.192 | 0.078 | 0.060 | **0.509** | 0.363 | 0.081 | 0.261 | 0.173 | 0.287 |
| pageblocks | 0.434 | 0.676 | 0.711 | 0.642 | 0.594 | 0.603 | 0.520 | **0.735** | 0.635 | 0.594 | 0.681 | 0.647 | 0.413 | 0.409 | 0.675 | 0.703 |
| pendigits | 0.588 | **0.970** | 0.785 | 0.518 | 0.386 | 0.117 | 0.093 | 0.146 | 0.334 | 0.391 | 0.664 | 0.353 | 0.113 | 0.483 | 0.522 | 0.409 |
| pima | 0.737 | 0.754 | 0.684 | 0.720 | 0.712 | 0.565 | 0.598 | 0.534 | 0.651 | 0.715 | **0.786** | 0.630 | 0.701 | 0.638 | 0.649 | 0.704 |
| satellite | 0.824 | 0.860 | 0.859 | 0.809 | 0.778 | 0.760 | 0.811 | 0.775 | 0.790 | 0.810 | 0.876 | **0.886** | 0.765 | 0.810 | 0.846 | 0.831 |
| satimage-2 | 0.945 | 0.967 | 0.885 | **0.969** | 0.919 | 0.475 | 0.763 | 0.793 | 0.959 | 0.929 | 0.947 | 0.954 | 0.967 | 0.866 | 0.714 | 0.963 |
| shuttle | 0.986 | 0.979 | **0.998** | 0.977 | 0.963 | 0.660 | 0.980 | 0.134 | 0.602 | 0.963 | 0.997 | 0.980 | 0.952 | 0.953 | 0.942 | 0.985 |
| skin | 0.646 | **0.982** | 0.617 | 0.663 | 0.364 | 0.504 | 0.430 | 0.656 | 0.422 | 0.401 | 0.325 | 0.787 | 0.723 | 0.797 | 0.655 | 0.603 |
| smtp | 0.011 | 0.505 | 0.481 | **0.645** | 0.495 | 0.209 | 0.307 | 0.087 | 0.324 | 0.494 | 0.038 | 0.500 | 0.473 | 0.594 | 0.440 | 0.547 |
| spambase | **0.883** | 0.833 | 0.727 | 0.822 | 0.818 | 0.742 | 0.753 | 0.791 | 0.821 | 0.818 | 0.868 | 0.856 | 0.807 | 0.784 | 0.838 | 0.831 |
| speech | 0.033 | 0.028 | 0.032 | 0.028 | 0.028 | 0.040 | 0.034 | 0.036 | 0.028 | 0.028 | 0.034 | 0.031 | 0.032 | 0.034 | 0.029 | **0.336** |
| stamps | 0.588 | 0.717 | 0.648 | 0.649 | 0.588 | 0.465 | 0.426 | 0.285 | 0.496 | 0.599 | **0.795** | 0.506 | 0.486 | 0.485 | 0.554 | 0.635 |
| thyroid | 0.797 | 0.809 | 0.606 | 0.789 | 0.813 | 0.631 | 0.691 | 0.744 | 0.801 | 0.813 | 0.515 | 0.741 | 0.585 | 0.406 | **0.832** | 0.818 |
| vertebral | 0.207 | 0.261 | 0.339 | 0.222 | 0.193 | 0.251 | 0.234 | 0.234 | 0.214 | 0.178 | 0.588 | 0.199 | 0.186 | 0.184 | 0.193 | **0.702** |
| vowels | 0.120 | 0.302 | 0.331 | 0.274 | 0.105 | 0.073 | 0.169 | 0.132 | 0.209 | 0.101 | 0.274 | **0.392** | 0.091 | 0.386 | 0.384 | 0.323 |
| waveform | 0.105 | 0.270 | **0.307** | 0.109 | 0.084 | 0.061 | 0.115 | 0.201 | 0.089 | 0.084 | 0.186 | 0.053 | 0.053 | 0.083 | 0.103 | 0.104 |
| wbc | 0.942 | 0.920 | 0.249 | 0.972 | 0.943 | 0.568 | 0.565 | 0.239 | 0.919 | 0.932 | 0.951 | **0.981** | 0.912 | 0.765 | 0.337 | 0.819 |
| wdbc | 0.720 | 0.820 | 0.936 | 0.874 | 0.821 | 0.309 | 0.843 | 0.122 | 0.788 | 0.836 | **0.956** | 0.891 | 0.903 | 0.903 | 0.714 | 0.944 |
| wilt | 0.088 | 0.122 | 0.157 | 0.071 | 0.064 | 0.084 | 0.071 | 0.096 | 0.109 | 0.072 | 0.289 | 0.122 | 0.077 | 0.237 | 0.253 | **0.313** |
| wine | 0.671 | 0.951 | 0.899 | 0.887 | 0.692 | 0.509 | 0.786 | 0.185 | 0.701 | 0.695 | 0.983 | **1.000** | 0.930 | 0.902 | 0.127 | 0.877 |
| wpbc | 0.407 | 0.461 | 0.412 | 0.409 | 0.400 | 0.372 | 0.749 | 0.360 | 0.389 | 0.403 | **0.893** | 0.875 | 0.412 | 0.365 | 0.401 | 0.838 |
| yeast | 0.468 | 0.483 | 0.489 | 0.479 | 0.468 | 0.492 | 0.492 | 0.498 | 0.508 | 0.465 | 0.495 | 0.506 | 0.467 | 0.499 | 0.500 | **0.537** |
| Average Value | 0.481 | 0.621 | 0.566 | 0.555 | 0.515 | 0.393 | 0.487 | 0.365 | 0.478 | 0.518 | 0.612 | 0.568 | 0.488 | 0.538 | 0.530 | **0.622** |
| Average Ranking | 9.46 | 5.23 | 7.35 | 7.02 | 9.66 | 12.45 | 10.30 | 11.53 | 10.52 | 9.39 | 5.64 | 6.94 | 9.70 | 8.15 | 7.87 | **4.79** |

**F1 scores of AGNI on full benchmark datasets** Based on the F1 scores (↑) summarized in Table A4, AGNI achieves the highest average F1 score and best average rank across 47 datasets, consistently demonstrating robustness across diverse evaluation metrics.

Table A4: Comparison of F1 scores (↑) across 47 tabular anomaly detection datasets. Each value represents the average over five independent runs with different random seeds. For each dataset, the best result is highlighted in **bold**, and the second-best is underlined.

| Dataset | IForest | kNN | LOF | OC-SVM | PCA | DA-GMM | Deep-SVDD | DRO-CC | GOAD | VAE | ICL | SLAD | MCM | DRL | DTE | AGNI |
|---|---|---|---|---|---|---|---|---|---|---|---|---|---|---|---|---|
| aloi | 0.042 | 0.059 | 0.082 | 0.073 | 0.076 | 0.060 | 0.052 | 0.000 | 0.057 | 0.076 | 0.049 | 0.053 | **0.099** | 0.090 | 0.043 | 0.072 |
| annthyroid | 0.550 | 0.620 | 0.496 | 0.536 | 0.500 | 0.457 | 0.233 | 0.574 | 0.558 | 0.502 | 0.494 | 0.660 | 0.342 | 0.260 | **0.773** | 0.715 |
| backdoor | 0.041 | 0.520 | 0.724 | 0.079 | 0.083 | 0.052 | 0.830 | 0.854 | 0.048 | 0.085 | 0.872 | 0.000 | 0.678 | **0.873** | 0.797 | 0.738 |
| breastw | 0.969 | 0.958 | 0.854 | 0.967 | 0.958 | 0.835 | 0.918 | 0.483 | 0.957 | 0.961 | 0.959 | 0.969 | **0.975** | 0.951 | 0.886 | 0.967 |
| campaign | 0.437 | 0.504 | 0.422 | 0.496 | 0.488 | 0.341 | 0.379 | 0.000 | 0.226 | 0.488 | 0.510 | 0.498 | 0.427 | 0.477 | 0.520 | **0.591** |
| cardio | 0.675 | 0.619 | 0.625 | 0.705 | 0.761 | 0.531 | 0.384 | 0.469 | 0.749 | 0.761 | 0.522 | 0.608 | 0.408 | 0.598 | 0.580 | **0.837** |
| cardiotocography | 0.561 | 0.464 | 0.483 | 0.579 | 0.616 | 0.523 | 0.371 | 0.340 | 0.600 | 0.616 | 0.389 | 0.338 | **0.690** | 0.587 | 0.406 | 0.630 |
| celeba | 0.173 | 0.172 | 0.019 | **0.274** | 0.272 | 0.142 | 0.084 | 0.086 | 0.040 | 0.270 | 0.127 | 0.137 | 0.102 | 0.159 | 0.199 | 0.244 |
| census | 0.105 | 0.225 | 0.131 | 0.207 | 0.208 | 0.145 | 0.193 | 0.155 | 0.050 | 0.208 | 0.240 | 0.086 | 0.154 | 0.095 | 0.179 | 0.228 |
| cover | 0.116 | 0.651 | **0.824** | 0.245 | 0.162 | 0.122 | 0.034 | 0.419 | 0.000 | 0.162 | 0.400 | 0.091 | 0.000 | 0.441 | 0.341 | 0.714 |
| donors | 0.435 | 0.949 | 0.745 | 0.395 | 0.373 | 0.219 | 0.414 | 0.294 | 0.043 | 0.378 | **0.972** | 0.559 | 0.645 | 0.805 | 0.813 | 0.407 |
| fault | 0.536 | 0.556 | 0.507 | 0.551 | 0.553 | 0.532 | 0.549 | 0.567 | 0.560 | 0.552 | 0.576 | **0.601** | 0.599 | 0.567 | 0.552 | 0.590 |
| fraud | 0.280 | 0.452 | 0.595 | 0.415 | 0.333 | 0.209 | 0.581 | 0.000 | 0.373 | 0.345 | 0.574 | 0.474 | 0.507 | 0.639 | **0.801** | 0.460 |
| glass | 0.162 | 0.259 | 0.205 | 0.150 | 0.158 | 0.137 | 0.454 | 0.155 | 0.202 | 0.180 | **0.878** | 0.350 | 0.150 | 0.200 | 0.178 | 0.340 |
| hepatitis | 0.540 | 0.813 | 0.419 | 0.666 | 0.606 | 0.475 | 0.938 | 0.293 | 0.579 | 0.605 | **0.996** | **0.996** | 0.298 | 0.329 | 0.508 | 0.529 |
| http | 0.258 | **1.000** | 0.968 | 0.998 | 0.927 | 0.489 | 0.250 | 0.000 | 0.564 | 0.919 | 0.607 | 0.885 | 0.259 | 0.990 | 0.019 | **1.000** |
| internetads | 0.264 | 0.519 | 0.546 | 0.462 | 0.457 | 0.319 | 0.543 | 0.384 | 0.461 | 0.457 | 0.559 | 0.578 | 0.577 | 0.440 | **0.641** | 0.471 |
| ionosphere | 0.834 | 0.905 | 0.875 | 0.926 | 0.790 | 0.693 | 0.931 | 0.602 | 0.834 | 0.798 | **0.942** | 0.926 | 0.611 | 0.901 | 0.890 | 0.939 |
| landsat | 0.433 | 0.515 | 0.536 | 0.386 | 0.340 | 0.409 | 0.422 | 0.408 | 0.330 | 0.388 | **0.538** | 0.469 | 0.385 | 0.517 | 0.379 | 0.439 |
| letter | 0.038 | 0.010 | 0.100 | 0.010 | 0.010 | 0.086 | 0.050 | **0.136** | 0.012 | 0.010 | 0.072 | 0.016 | 0.024 | 0.056 | 0.026 | 0.020 |
| lymphography | 0.851 | 0.945 | 0.749 | **1.000** | 0.909 | 0.676 | 0.898 | 0.262 | 0.931 | 0.930 | **1.000** | 0.995 | 0.400 | 0.267 | **1.000** | **1.000** |
| magic.gamma | 0.696 | 0.762 | 0.761 | 0.684 | 0.652 | 0.574 | 0.599 | 0.726 | 0.627 | 0.652 | 0.696 | 0.659 | 0.762 | 0.744 | **0.807** | 0.696 |
| mammography | 0.392 | 0.404 | 0.385 | 0.419 | 0.446 | 0.269 | 0.316 | 0.327 | 0.356 | 0.450 | 0.174 | 0.222 | 0.329 | **0.536** | 0.375 | 0.452 |
| mnist | 0.526 | 0.719 | 0.714 | 0.643 | 0.639 | 0.447 | 0.433 | 0.573 | 0.639 | 0.639 | 0.649 | 0.670 | 0.743 | **0.790** | 0.605 | 0.755 |
| musk | 0.359 | **1.000** | **1.000** | **1.000** | **1.000** | 0.707 | 0.992 | 0.122 | **1.000** | **1.000** | 0.833 | **1.000** | 0.522 | 0.380 | **1.000** | 0.990 |
| optdigits | 0.128 | 0.213 | 0.533 | 0.007 | 0.007 | 0.003 | 0.000 | 0.201 | 0.004 | 0.007 | **0.577** | 0.399 | 0.045 | 0.292 | 0.136 | 0.347 |
| pageblocks | 0.426 | 0.590 | 0.659 | 0.557 | 0.469 | 0.579 | 0.547 | **0.684** | 0.502 | 0.469 | 0.649 | 0.602 | 0.378 | 0.403 | 0.629 | 0.632 |
| pendigits | 0.580 | **0.904** | 0.763 | 0.532 | 0.442 | 0.140 | 0.123 | 0.192 | 0.415 | 0.442 | 0.612 | 0.444 | 0.106 | 0.471 | 0.624 | 0.458 |
| pima | 0.696 | 0.706 | 0.667 | 0.686 | 0.693 | 0.540 | 0.559 | 0.500 | 0.592 | 0.705 | **0.735** | 0.589 | 0.681 | 0.608 | 0.625 | 0.702 |
| satellite | 0.671 | 0.718 | 0.726 | 0.673 | 0.627 | 0.651 | 0.678 | 0.675 | 0.636 | 0.662 | 0.750 | **0.782** | 0.596 | 0.674 | 0.721 | 0.695 |
| satimage-2 | 0.896 | 0.901 | 0.817 | **0.915** | 0.873 | 0.504 | 0.732 | 0.763 | 0.907 | 0.882 | 0.884 | 0.887 | 0.910 | 0.839 | 0.694 | 0.907 |
| shuttle | 0.967 | 0.982 | 0.984 | 0.965 | 0.958 | 0.679 | 0.981 | 0.000 | 0.563 | 0.958 | **0.988** | 0.985 | 0.977 | 0.956 | 0.980 | 0.983 |
| skin | 0.781 | **0.964** | 0.708 | 0.800 | 0.379 | 0.557 | 0.433 | 0.784 | 0.520 | 0.447 | 0.011 | 0.746 | 0.750 | 0.835 | 0.784 | 0.698 |
| smtp | 0.000 | 0.695 | 0.658 | 0.695 | 0.695 | 0.263 | 0.340 | 0.138 | 0.486 | **0.696** | 0.070 | **0.696** | 0.462 | 0.664 | 0.667 | 0.667 |
| spambase | 0.805 | 0.805 | 0.740 | 0.786 | 0.785 | 0.684 | 0.696 | 0.739 | 0.788 | 0.785 | 0.793 | **0.815** | 0.767 | 0.718 | 0.800 | 0.814 |
| speech | 0.039 | 0.033 | 0.033 | 0.033 | 0.033 | 0.033 | 0.013 | 0.029 | 0.029 | 0.033 | 0.026 | 0.062 | 0.039 | 0.026 | 0.043 | **0.326** |
| stamps | 0.636 | 0.755 | 0.635 | 0.634 | 0.579 | 0.470 | 0.371 | 0.280 | 0.527 | 0.614 | **0.772** | 0.510 | 0.512 | 0.437 | 0.529 | 0.650 |
| thyroid | **0.804** | 0.753 | 0.527 | 0.753 | 0.742 | 0.654 | 0.656 | 0.690 | 0.742 | 0.742 | 0.561 | 0.712 | 0.551 | 0.419 | 0.768 | 0.762 |
| vertebral | 0.158 | 0.238 | 0.337 | 0.204 | 0.139 | 0.212 | 0.167 | 0.170 | 0.182 | 0.141 | **0.634** | 0.142 | 0.080 | 0.093 | 0.120 | 0.627 |
| vowels | 0.152 | 0.260 | 0.340 | 0.280 | 0.120 | 0.056 | 0.208 | 0.136 | 0.236 | 0.120 | 0.244 | 0.388 | 0.036 | **0.396** | 0.392 | 0.320 |
| waveform | 0.102 | 0.270 | **0.280** | 0.130 | 0.090 | 0.046 | 0.146 | 0.266 | 0.098 | 0.080 | 0.268 | 0.022 | 0.063 | 0.077 | 0.126 | 0.111 |
| wbc | 0.882 | 0.864 | 0.203 | 0.898 | 0.873 | 0.462 | 0.542 | 0.266 | 0.865 | 0.884 | **0.929** | 0.922 | 0.909 | 0.709 | 0.360 | 0.745 |
| wdbc | 0.709 | 0.787 | 0.856 | 0.803 | 0.788 | 0.325 | 0.833 | 0.087 | 0.758 | 0.787 | 0.905 | 0.852 | **0.909** | 0.818 | 0.620 | 0.818 |
| wilt | 0.020 | 0.023 | 0.167 | 0.012 | 0.016 | 0.057 | 0.006 | 0.015 | 0.124 | 0.019 | **0.352** | 0.070 | 0.000 | 0.321 | 0.163 | 0.288 |
| wine | 0.711 | 0.872 | 0.808 | 0.783 | 0.660 | 0.485 | 0.698 | 0.128 | 0.655 | 0.679 | 0.993 | **1.000** | 0.909 | 0.800 | 0.000 | 0.800 |
| wpbc | 0.366 | 0.491 | 0.413 | 0.358 | 0.336 | 0.332 | 0.702 | 0.319 | 0.342 | 0.365 | **0.905** | 0.879 | 0.396 | 0.329 | 0.396 | 0.838 |
| yeast | 0.445 | 0.468 | 0.477 | 0.466 | 0.434 | 0.520 | 0.495 | 0.488 | **0.532** | 0.443 | 0.503 | 0.493 | 0.462 | 0.492 | 0.497 | 0.521 |
| Average Value | 0.452 | 0.593 | 0.555 | 0.528 | 0.490 | 0.377 | 0.463 | 0.336 | 0.453 | 0.498 | 0.591 | 0.550 | 0.452 | 0.512 | 0.510 | **0.607** |
| Average Ranking | 9.40 | 5.67 | 7.15 | 7.54 | 9.61 | 12.35 | 10.51 | 11.66 | 10.24 | 9.00 | 5.66 | 6.80 | 9.59 | 8.48 | 7.57 | **4.77** |

Table A5: Detailed results of the component-wise experiments on the subset

| Setting | Performance | annthyroid | cardiotocography | stamps | vowels | yeast | Avg. |
|---------|-------------|------------|------------------|--------|--------|-------|------|
| A | AUC-ROC | 0.957 | 0.788 | 0.920 | 0.668 | 0.506 | 0.768 |
|   | AUC-PR | 0.752 | 0.699 | 0.583 | 0.216 | 0.525 | 0.555 |
|   | F1 score | 0.697 | 0.631 | 0.625 | 0.220 | 0.518 | 0.538 |
|   | Time (s) | 0.030 | 0.019 | 0.022 | 0.019 | 0.018 | 0.022 |
| B | AUC-ROC | 0.958 | 0.793 | 0.913 | 0.622 | 0.535 | 0.764 |
|   | AUC-PR | 0.759 | 0.713 | 0.573 | 0.204 | 0.545 | 0.559 |
|   | F1 score | 0.720 | 0.632 | 0.625 | 0.297 | 0.526 | 0.560 |
|   | Time (s) | 0.030 | 0.019 | 0.023 | 0.019 | 0.020 | 0.022 |
| C | AUC-ROC | 0.955 | 0.804 | 0.936 | 0.721 | 0.535 | 0.790 |
|   | AUC-PR | 0.775 | 0.717 | 0.630 | 0.228 | 0.539 | 0.578 |
|   | F1 score | 0.715 | 0.624 | 0.656 | 0.247 | 0.531 | 0.555 |
|   | Time (s) | 0.044 | 0.030 | 0.030 | 0.027 | 0.027 | 0.031 |
| D (AGNI) | AUC-ROC | 0.974 | 0.826 | 0.930 | 0.754 | 0.553 | **0.808** |
|   | AUC-PR | 0.813 | 0.760 | 0.649 | 0.301 | 0.548 | **0.614** |
|   | F1 score | 0.766 | 0.687 | 0.688 | 0.300 | 0.543 | **0.597** |
|   | Time (s) | 0.044 | 0.029 | 0.030 | 0.030 | 0.027 | 0.032 |

**Detailed results of component-wise analysis**  Table A5 reports the component-wise ablation results of AGNI on the selected subset (*annthyroid*, *cardiotocography*, *stamps*, *vowels*, and *yeast*)). Setting A-D correspond to the configuration described in Table 4 of the main text and are detailed as follows. All experiments were conducted using a single random seed.

- Setting A: A vanilla transformer-based autoencoder without any of the proposed components.

- Setting B: Incorporates the attention-guided masking component into Setting A. The masking ratio is varied from 0.3 to 0.8 in increments of 0.1, and the average performance is reported.

- Setting C: Incorporates the neighbor-informed reconstruction component info Setting A. The number of neighbors is set to 1, 2, and 3, and results are averaged across these values.

- Setting D: Represents the full AGNI model, which integrates both proposed components. For each dataset, the masking ratio and number of neighbors are set to the best-performing configuration.

**Detailed results of configuration-wise variants on the subset**  Table A6 presents the performance of five configuration-wise variants evaluated on the five subsets. Each variant modifies one or more components of the masking strategy, the neighbor selection criterion, and the fusion strategy. Variant 5 corresponds to AGNI, our proposed framework. AGNI (Variant 5) achieves the best average performance in all evaluation metrics, demonstrating the effectiveness of its design. A closer comparison with other variants highlights the contribution of each component:

- **Attention-guided masking vs. random masking (Variant 5 vs. Variant 4):** Replacing attention-guided masking with random masking leads to performance drops, particularly in AUC-PR (from 0.614 to 0.483) and F1 score (from 0.597 to 0.501). This indicates that masking based on attention scores allows more informative and discriminative learning.

- **Attention-based neighbor selection vs. embedding similarity (Variant 5 vs. Variant 3):** Using embedding similarity for neighbor selection instead of attention scores results in lower performance across all metrics. This suggests that attention-based selection captures structural relationships more effectively.

- **Fusion by individual concatenation vs. averaging (Variant 5 vs. Variants 1 and 2):** Variants 1 and 2 use averaging-based fusion strategies, which underperform compared to direct concatenation. This shows that reserving neighbor-specific signals through explicit concatenation leads to better utilization of contextual information.

Overall, these results validate the effectiveness of AGNI's configuration and demonstrate that each component—attention-guided masking, attention-based neighbor selection, and concatenation-based fusion—contributes meaningfully to performance improvements.

Table A6: Detailed results of Evaluation results for configuration-wise variants on the subset.

| Variant | Masking | Neighbor-selection criteria | Fusion strategy | Metric | annthyroid | cardio-tocography | stamps | vowels | yeast | Avg. |
|---|---|---|---|---|---|---|---|---|---|---|
| 1 | Attention-guided | Attention score | $\left[\frac{1}{M+1}\left(z + \sum_i^M \mathbf{z}_i\right)\right]$ | AUC-ROC | 0.873 | 0.784 | 0.916 | 0.825 | 0.443 | 0.768 |
| | | | | AUC-PR | 0.585 | 0.694 | 0.561 | 0.233 | 0.472 | 0.509 |
| | | | | F1 score | 0.515 | 0.612 | 0.563 | 0.280 | 0.465 | 0.487 |
| 2 | Attention-guided | Attention score | $[z; \frac{1}{M}\sum_i^M \mathbf{z}_i]$ | AUC-ROC | 0.977 | 0.793 | 0.908 | 0.716 | 0.561 | 0.791 |
| | | | | AUC-PR | 0.827 | 0.707 | 0.550 | 0.249 | 0.555 | 0.578 |
| | | | | F1 score | 0.800 | 0.620 | 0.563 | 0.300 | 0.549 | 0.566 |
| 3 | Attention-guided | Embedding sim. | $[z; z_1; \ldots; z_M]$ | AUC-ROC | 0.966 | 0.784 | 0.914 | 0.826 | 0.513 | 0.801 |
| | | | | AUC-PR | 0.769 | 0.693 | 0.557 | 0.325 | 0.529 | 0.575 |
| | | | | F1 score | 0.719 | 0.616 | 0.563 | 0.340 | 0.514 | 0.550 |
| 4 | Random | Attention score | $[z; z_1; \ldots; z_M]$ | AUC-ROC | 0.924 | 0.726 | 0.773 | 0.616 | 0.519 | 0.712 |
| | | | | AUC-PR | 0.743 | 0.617 | 0.354 | 0.171 | 0.531 | 0.483 |
| | | | | F1 score | 0.723 | 0.552 | 0.438 | 0.260 | 0.531 | 0.501 |
| 5 (AGNI) | Attention-guided | Attention score | $[z; z_1; \ldots; z_M]$ | AUC-ROC | 0.974 | 0.826 | 0.930 | 0.754 | 0.553 | **0.808** |
| | | | | AUC-PR | 0.813 | 0.760 | 0.649 | 0.301 | 0.548 | **0.614** |
| | | | | F1 score | 0.766 | 0.687 | 0.688 | 0.300 | 0.543 | **0.597** |

Table A7: Optimal masking ratio and number of neighbors per dataset.

| Dataset | Masking Ratio | Neighbors | Dataset | Masking Ratio | Neighbors |
|---|---|---|---|---|---|
| aloi | 0.7 | 1 | musk | 0.4 | 3 |
| annthyroid | 0.3 | 1 | optdigits | 0.7 | 1 |
| backdoor | 0.4 | 2 | pageblocks | 0.6 | 1 |
| breastw | 0.4 | 3 | pendigits | 0.4 | 3 |
| campaign | 0.6 | 1 | pima | 0.3 | 2 |
| cardio | 0.4 | 1 | satellite | 0.4 | 2 |
| cardiotocography | 0.5 | 1 | satimage-2 | 0.4 | 3 |
| celeba | 0.8 | 2 | shuttle | 0.5 | 1 |
| census | 0.5 | 3 | skin | 0.7 | 3 |
| cover | 0.6 | 1 | smtp | 0.7 | 2 |
| donors | 0.3 | 1 | spambase | 0.7 | 2 |
| fault | 0.3 | 3 | speech | 0.3 | 2 |
| fraud | 0.7 | 3 | stamps | 0.6 | 1 |
| glass | 0.8 | 1 | thyroid | 0.7 | 1 |
| hepatitis | 0.3 | 2 | vertebral | 0.4 | 1 |
| http | 0.8 | 3 | vowels | 0.7 | 3 |
| internetads | 0.4 | 3 | waveform | 0.8 | 3 |
| ionosphere | 0.8 | 3 | wbc | 0.4 | 3 |
| landsat | 0.6 | 1 | wdbc | 0.3 | 1 |
| letter | 0.4 | 2 | wilt | 0.6 | 1 |
| lymphography | 0.3 | 1 | wine | 0.5 | 1 |
| magic | 0.8 | 1 | wpbc | 0.6 | 3 |
| mammography | 0.4 | 1 | yeast | 0.7 | 2 |
| mnist | 0.5 | 3 | | | |

**Anomaly Ratio Analysis** At test time, the ratio of anomalies to normal samples can affect evaluation results, but AGNI's detection mechanism—based on reconstruction errors and learned normal patterns—remains stable across different anomaly proportions. To empirically validate this robustness, we leveraged the ADBench benchmark used in our study, which comprises 47 datasets. We categorized these datasets into three groups based on their anomaly ratios, with each category containing a similar number of datasets: low ratio (< 3.2%), medium ratio (3.2-10.2%), and high ratio

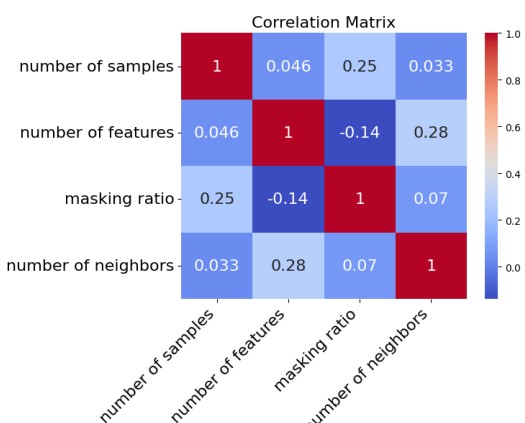

Figure A1: Spearman correlation heatmap of 47 real-world tabular datasets

Table A8: AUC-ROC performance by anomaly ratio groups (mean±std)

| Method | Low (0, 3.2] | Medium (3.2, 10.2] | High (10.2, 40] |
|---|---|---|---|
| IForest | 0.832±0.181 | 0.809±0.205 | 0.707±0.176 |
| kNN | 0.871±0.190 | 0.872±0.178 | 0.762±0.167 |
| LOF | 0.851±0.217 | 0.853±0.160 | 0.708±0.138 |
| OCSVM | 0.829±0.198 | 0.819±0.221 | 0.719±0.174 |
| PCA | 0.819±0.201 | 0.789±0.243 | 0.676±0.172 |
| DAGMM | 0.710±0.183 | 0.681±0.176 | 0.621±0.116 |
| DeepSVDD | 0.703±0.210 | 0.704±0.209 | 0.683±0.180 |
| DROCC | 0.683±0.219 | 0.601±0.175 | 0.575±0.141 |
| GOAD | 0.706±0.262 | 0.719±0.246 | 0.664±0.168 |
| VAE | 0.819±0.201 | 0.796±0.230 | 0.694±0.163 |
| ICL | 0.841±0.178 | 0.873±0.159 | 0.741±0.239 |
| SLAD | 0.799±0.217 | 0.834±0.181 | 0.757±0.191 |
| MCM | 0.770±0.214 | 0.765±0.197 | 0.666±0.180 |
| DRL | 0.835±0.167 | 0.810±0.174 | 0.695±0.177 |
| DTE | 0.856±0.185 | 0.824±0.196 | 0.713±0.173 |
| AGNI | **0.908±0.127** | **0.878±0.155** | **0.790±0.133** |

(> 10.2%). Table A8 shows the analysis results. Analysis of AUC-ROC for each group revealed that AGNI achieved the highest AUC-ROC performance across all three categories while maintaining the smallest standard deviations. These experimental findings confirm that AGNI maintains superior and consistent detection performance across a wide range of conditions from low to high anomaly ratios. We attribute this robustness to AGNI's adaptive design: although a higher anomaly ratio increases the likelihood of including anomalous samples among the retrieved neighbors, AGNI's attention mechanism assigns low weights to such outliers, thereby limiting their influence on reconstruction. Additionally, since anomaly scores are based on instance-specific reconstruction errors of informative features, even when neighbors are noisy, abnormal instances still yield significantly larger errors, enabling robust detection via relative ranking.

**Dataset-wise analysis of optimal masking ratio and neighbor count**  We report the optimal masking ratio and number of neighbors per dataset in Table A7. According to the Spearman correlation heatmap results Figure A1, a weak positive correlation ($\rho \approx 0.25$) was observed between masking ratio and sample size. This suggests that datasets with larger sample sizes tended to benefit from relatively higher masking ratios for performance improvement. Similarly, a positive correlation of comparable magnitude ($\rho \approx 0.28$) was found between the number of features and the number of neighbors, indicating that configurations using more neighbors were more frequently selected for optimal performance in cases with higher feature dimensionality.

---

**Algorithm 1** Attention-Guided Masking and Neighbor-Aware Decoding

---

**Require:** Training data $\mathcal{D}_{\text{train}}$, masking ratio $\rho$, neighbors $K$
**Ensure:** Trained encoder $f_{\text{enc}}$ and decoder $f_{\text{dec}}$
  Initialize encoder $f_{\text{enc}}$ and decoder $f_{\text{dec}}$
  **repeat**
    **for** each mini-batch $\mathcal{B} \subset \mathcal{D}_{\text{train}}$ **do**
      Compute attention scores $\{\mathbf{s}_i\}$ for all $\mathbf{x}_i \in \mathcal{B}$
      $\mathcal{L}_{\text{batch}} \leftarrow 0$
      **for** each sample $\mathbf{x}_i \in \mathcal{B}$ **do**
        Mask top-$\rho$ features based on $\mathbf{s}_i$ to get $\tilde{\mathbf{x}}_i$
        Find $K$ neighbors with most similar attention score vectors
        $\mathbf{z}_i \leftarrow f_{\text{enc}}(\tilde{\mathbf{x}}_i)$
        $\mathbf{Z}_{\text{neighbors}} \leftarrow$ encoded representations of $K$ neighbors
        $\mathbf{c}_i \leftarrow \text{Concatenate}(\mathbf{z}_i, \mathbf{Z}_{\text{neighbors}})$
        $\hat{\mathbf{x}}_i \leftarrow f_{\text{dec}}(\mathbf{c}_i)$
        $\mathcal{L}_{\text{batch}} \leftarrow \mathcal{L}_{\text{batch}} + \|\mathbf{x}_i - \hat{\mathbf{x}}_i\|_2^2$
      **end for**
      Update model parameters using $\mathcal{L}_{\text{batch}}/|\mathcal{B}|$
    **end for**
  **until** convergence

---

## D  ALGORITHM

For completeness, we present the pseudocode of our training procedure in Algorithm 1. The algorithm summarizes the two key components described in method section: (1) attention-guided feature masking based on transformer attention scores, and (2) neighbor-informed reconstruction using representations of attention-similar samples. This formulation highlights how instance-specific masking and contextual reconstruction are integrated during self-supervised training.

## E  COMPUTATION AND RUNTIME ANALYSIS

### E.1  STEP-WISE COMPUTATION ANALYSIS

We analyze the test-time computational complexity of our method in terms of the batch size $B$, input feature dimension $d$, and number of attention heads $H$. We also assume that the number of neighbors $M$ is a small constant (we use $M = 3$ in all experiments) and is therefore omitted from asymptotic expressions.

**Step 1: Transformer encoding and attention extraction.** Each of the $B$ test samples is passed through a transformer encoder. The cost of multi-head self-attention is $O(Hd^2)$ per sample, resulting in:

$$O(BHd^2)$$

Attention scores are extracted from the final encoder layer and reduced via max pooling over heads and features, which adds only $O(BHd)$ cost—negligible compared to the encoder's complexity.

**Step 2: Masking.** Generating and applying binary masks involves sorting or thresholding the $d$-dimensional attention score vector per sample. This step requires $O(Bd)$ operations and is dominated by the encoder and retrieval costs.

**Step 3: Pairwise similarity computation.** We compute cosine similarity between the attention score vectors for all sample pairs within the batch. Each similarity computation takes $O(d)$ time, resulting in:

$$O(B^2d)$$

**Step 4: Neighbor retrieval.** Each sample retrieves its top-$M$ nearest neighbors based on cosine similarity. Top-$M$ selection can be done in $O(B)$ time per sample, yielding:

$$O(B^2)$$

This is strictly dominated by the previous step and thus omitted from the final complexity expression.

**Step 5: MLP decoding.** For each sample, the decoder receives a concatenated representation of its own latent vector and those of its $M$ retrieved neighbors, producing a vector of dimension $(M+1){\cdot}d$. The decoder is a 3-layer MLP with fully connected layers of hidden size $d$, yielding a per-sample cost of $O((M+1)^2d^2) = O(d^2)$. Across all $B$ samples:

$$O(Bd^2)$$

**Total complexity.** Combining all dominant steps, the overall test-time complexity is:

$$O(BHd^2 + B^2d)$$

In typical settings (e.g., $H = 8$, $d = 100$, $B = 64$), the $B^2d$ term arising from similarity computation dominates. This justifies our use of moderate batch sizes, which provide a favorable trade-off between retrieval quality and computational efficiency.

