# OpenReview forum: "Attention-Guided Masking and Neighbor-Informed Reconstruction for Tabular Anomaly Detection"
_ICLR.cc/2026/Conference — ICLR 2026 Conference Withdrawn Submission_

### Official Review · Reviewer_R1oB · 2025-10-16

**Soundness:** 4
**Presentation:** 3
**Contribution:** 2
**Rating:** 6
**Confidence:** 5

**Summary:**

The present work proposes a novel anomaly detection (AD) method in the weakly-supervised/OCC setting. The latter consists in disposing of a labeled dataset comprised of *normal* and *anomaly* samples. The learning tasks consists in characterizing the normal distribution so that in inference samples can be predicted to stem from the estimated *normal* distribution or be considered as anomalies.

The proposed method, AGNI, is a self-supervised approach to AD that relies on mask reconstruction as pretext task. AGNI uses as anomaly score the same metric as the optimized one during training: the reconstructions score. Author claim that the novelty of their work lies in two elements:
- they combine **inter-sample** and **inter-feature** (or intra-sample) dependencies, by relying on retrieval-augmented learning. In short they use a retrieval attention-based module to identify relevant neighbors that can help the model form its prediction. The latter module helps capture the inter-sample dependencies. In parallel, the intra-sample dependencies are inherently captured by the transformer encoder and MLPs.
- they rely on an attention guided masking strategy to select the most relevant mask during training and inference.

The experiments conducted on a widely used AD benchmark, ADBench, supports the relevance of the method as they demonstrate performance surpassing existing methods to which they compared.

**Strengths:**

**S1**: The paper is well-written, well-structured and easy to follow.

**S2**: The experiments are rigorous and thorough. The ablation studies are rather complete and answer most of the questions one can have regarding the novel components tested in their method.

**S3**: Experiments are partially reproductible as the authors make the raw training code available in the supplementary material.

**Weaknesses:**

**W1**: Some **key references are missing** from the paper (e.g. [1], [2] and [3]). The authors do not refer to [1] and [2], while they are **highly relevant given the authors claims**. In particular, given our understanding of AGNI and NPT-AD [1], it seems that the claim from the author (line 55) that "the intra-instance and inter-sample perpsective (...) remain unexplored in tabular anomaly detection" is imprecise.

In particular, [1] explictely motivates their approach by the fact that no existing AD method has leveraged both types of dependencies. This is the exact same claim as the authors for the present work. Moreover, it seems that [1] also relies on a mask reconstruction SSL approach to train their AD, making their method very similar to the present work. Additionally, their retrieval method is also attention-based as it explicitely relies on the Non-Parametric Transformer's [2] ability to leverage inter-datapoints dependencies through their ABD layers. This somewhat mitigates the novelty of the proposed work.

Similarly, some recent method that have experimented on the same benchmark have been included but *wrongly attributed* as the authors attribute DRL [3] to [4] while [4] proposes a method coined Disent-AD. We urge the authors to carefuly check their citations, and to correct them. We recommend the authors to **include DRL** (correctly cited) in their benchmark, **NPT-AD** (whose performance on the ADBench benchmark is given in the DRL paper) and **Disent-AD** that has also experimented on the ADBench benchmark.

**W2**: Some important details should be included in the main text of the paper regarding training settings. (see the question section)



[1] Beyond Individual Input for Deep Anomaly Detection on Tabular Data. *H. Thimonier, F. Popineau, A. Rimmel, BL. Doan*. ICML, 2024.

[2] Self-Attention Between Datapoints: Going Beyond Individual Input-Output Pairs in Deep Learning. *Jannik Kossen, Neil Band, Clare Lyle, Aidan N. Gomez, Tom Rainforth, Yarin Gal*. NeurIPS 2021.

[3] DRL: Decomposed Representation Learning for Tabular Anomaly Detection. *Hangting Ye, He Zhao, Wei Fan, Mingyuan Zhou, Dan dan Guo, Yi Chang*. ICLR 2025.

[4] Disentangling Tabular Data Towards Better One-Class Anomaly Detection. *Jianan Ye, Zhaorui Tan, Yijie Hu, Xi Yang, Guangliang Cheng, Kaizhu Huang*. AAAI 2025.

**Questions:**

**Q1**: In line 260 the authors mention that **the number of tokens is set to 24**. Given our understanding of how transformer have been applied to tabular data in previous work, the number of tokens should be either the same as the number of features, or more because some additional tokens have been included (e.g. [CLS] as in [5]). In short each feature is embedded using some embedding module, and then each embedded features acts like a token in the sequence that serves as input to the transformer. Could you elaborate on this ?

**Q2**: While attention-guided masking is a clever trick, I am wondering how you (i) **initialize the mask strategy** for the first training step? Given my understanding, the mask at step t, is updated given the attention weights at step t-1. What happens for step 0?
Additionally, it seems that in the early stage of training the selected mask by your method could be erratic. Have you experienced an equivalent of "representation collapse" where the model for example alternates between two mask after each update, or situation where the model struggles to converge?
**It would be interesting to investigate mask variations as training evolves.**

**Q3**: Authors mention in line 350 that after M=3 (the retrieved neighbors) the gain seem to plateau, yet, the figure displays the performance for values between 1 and 3. The authors might consider including larger values of M to show this plateau-ing.

**Q4**: As discussed in [6], T-SNE suffers from large variability of the provided representation when varying the hyperparameters. Could the authors share the parameters used to provide the representation in Figure 4? Also, other dimension reduction technics might be interesting to include (e.g. UMAP).

Overall, I lean towards acceptance and would be happy to increase my score should my interrogations and **W1** be addressed.



[5] Revisiting Deep Learning Models for Tabular Data. *Yury Gorishniy, Ivan Rubachev, Valentin Khrulkov, Artem Babenko*. NeurIPS 2021.

[6] How to Use t-SNE Effectively", *Wattenberg, et al.*, "Distill, 2016. http://doi.org/10.23915/distill.00002

---

### Official Review · Reviewer_6v8Y · 2025-10-23

**Soundness:** 1
**Presentation:** 3
**Contribution:** 2
**Rating:** 0
**Confidence:** 4

**Summary:**

A masking-based method for anomaly detection in tabular data that uses the attention scores twice (to decide which features to mask and to find the nearest samples in the batch).

**Strengths:**

As a story telling the reader why the authors do what they do and how it is compared to the cited work, the paper is interesting, easy to read, and convincing.

I liked the idea of using the attention scores to find neighborhoods.

**Weaknesses:**

It is important that every test sample is evaluated independently and not as part of a batch of test samples. Since this was not clarified in the paper, I reviewed the code, and indeed, there is transductive learning taking place. This is an unfair advantage and a clear reason for rejection.

I also find it very problematic that, while a known benchmark is used, the authors rerun the baselines and also apply the benchmark datasets using their own protocol.

Figure 2 does not really tell us if the results are significant.

Figure 3 would benefit from having zero neighbors as well (this is given in Table 3 for the entire benchmark)

figure 3 and table 2 need to be executed on more than a handful of datasets.

Table 3 -- no real difference between setting A and B. This goes against the paper's narrative.

Table 4 variant 3 is not much worse than the final method, again, going against the paper's narrative.

Figure 4 is not very telling without baseline methods to compare to

The code has the notion of permutation to boost results that is NOT mentioned in the paper

**Questions:**

nothing to add

---

### Official Review · Reviewer_gjpH · 2025-10-27

**Soundness:** 2
**Presentation:** 3
**Contribution:** 3
**Rating:** 4
**Confidence:** 4

**Summary:**

This paper introduces AGNI (Attention-Guided Masking and Neighbor-Informed Reconstruction), a self-supervised framework for one-class classification in tabular anomaly detection. The key innovation is the dual use of attention mechanisms: (1) to identify and mask salient features for intra-instance modeling, and (2) to retrieve structurally similar neighbors for inter-sample contextual reconstruction. The authors evaluate AGNI on 47 datasets from the ADBench benchmark, demonstrating superior performance compared to 15 baseline methods.

**Strengths:**

1. Novel Unified Framework: The use of attention for both masking and retrieval is elegant and well-motivated. Unlike prior work that treats attention solely as a representational tool, AGNI transforms it into a coordinating structural signal.

2. Comprehensive Experimental Evaluation: The proposed method is evaluated on 47 diverse datasets with 15 baselines, and achieves the best performance across three metrics (AUC-ROC, AUC-PR, F1).

3. Qualitative Insights: The t-SNE visualizations (Figure 4) showing compact neighborhoods for normal samples versus scattered neighbors for anomalies provide valuable intuition about the method's behavior.

**Weaknesses:**

1. **Fundamental Flaw in Similarity Computation**: The GetAtt module in Equation 3 reduces the full attention matrix to a single scalar per feature by taking the maximum across all heads and positions: $s_i = \max_{h,j} a^h_{j,i}$. This aggressive reduction discards crucial structural information about *which* features attend to each other. Two samples can have identical attention score vectors but fundamentally different attention patterns. For example:
- Sample A: Feature $i$ receives maximum attention from feature $j$
- Sample B: Feature $i$ receives maximum attention from feature $k$ (where $j \neq k$)

Both yield the same $s_i$, yet their structural relationships are entirely different. Computing cosine similarity on these reduced scores  therefore cannot reliably identify structurally similar samples.

2. **Strong Dependence on Batch Composition**: Retrieval is performed within each batch, introducing a strong dependence on batch composition. Although the authors provide parameter sensitivity analysis for batch size, the batch-dependent neighbor retrieval fundamentally limits scalability. This design choice becomes particularly problematic for small batches, single-sample inference scenarios, or batches dominated by anomalies.

3. Limited discussion of failure cases: The paper doesn't discuss when AGNI performs poorly. For example, AGNI underperforms on some datasets like "donors" (0.885 vs 0.995 for kNN). Supplementing this discussion would clarify AGNI’s limitations, guide its future optimization, and help readers assess its practical applicability.

4. Lack of Symbol Explanations: Some symbols in the formulas lack necessary definitions and explanations, making it difficult for readers to accurately understand the meaning of the formulas. For example, the vector $s_i$ in Formula (5), and the symbols $z_i$ and $z_{i,j}$ in Formula (6) need clearly explained in the paper.

**Questions:**

See weakness

---

### Official Review · Reviewer_QL6A · 2025-10-31

**Soundness:** 2
**Presentation:** 3
**Contribution:** 2
**Rating:** 2
**Confidence:** 5

**Summary:**

The proposed AGNI framework presents a novel approach that couples intra-instance and inter-sample modeling through an attention-guided mechanism. Its strong performance across numerous datasets demonstrates significant practical value.

**Strengths:**

The method is intuitively sound. By masking important features, it forces the model to learn feature dependencies, and by incorporating neighbours, it introduces valuable contextual information. The core contribution lies in its innovative framework design and the unification of the dual roles of attention, rather than in theoretical novelty.

**Weaknesses:**

1. The paper does not provide a theoretical derivation and primarily focuses on the description of methods and experimental validation.
a.  The paper primarily offers methodological description and experimental validation, without providing theoretical derivations.
b. There is no formal proof or theoretical analysis to demonstrate why inter-sample modelling provides a superior advantage over intra-instance modelling, or how AGNI provably improves the modelling of the normal data distribution for enhanced anomaly detection.

2. Insufficient Justification for Attention as a Similarity Metric
a.  The rationale for using attention as a similarity measure is insufficiently substantiated. Why do attention scores reflect structural similarity more accurately than latent embeddings? There is a lack of theoretical or visual support for this claim.
b. The definition of "structural similarity" relies on attention scores; however, attention scores are outcomes of model training, which might lead to a circular dependency issue.

3. As analysed in Section E.1, the quadratic scaling with batch size can become a computational bottleneck for large batch sizes or high-dimensional data, limiting the method's scalability in real-world applications.

**Questions:**

Please prepare a rebuttal addressing the identified weaknesses.
In particular, consider how you can offer the readers some insightful ideas that could enhance their understanding or spark further exploration in the field.

---

### Note · Authors · 2025-11-18

**Comment:**

Dear Reviewers,

Thank you for taking the time to evaluate our manuscript. From your reviews, we realized that some of our main ideas and contributions were not fully understood. This shows us that we need to present our motivation and methodology more clearly. We will revise the manuscript to better highlight the core ideas and assumptions, reduce potential misunderstandings, and further refine the method, experiments, and overall narrative.

Best regards,
The authors

**Withdrawal Confirmation:**

I have read and agree with the venue's withdrawal policy on behalf of myself and my co-authors.